# MedCL: Learning Consistent Anatomy Distribution for Scribble-supervised Medical Image Segmentation

**Ke Zhang**                          KZHANG99@JHU.EDU

**Vishal M. Patel**                         VPATEL36@JHU.EDU

*Department of Electrical and Computer Engineering, Johns Hopkins University, USA*

**Editors:** Accepted for publication at MIDL 2025

## Abstract

Curating large-scale fully annotated datasets is expensive, laborious, and cumbersome, especially for medical images. Several methods have been proposed in the literature that make use of weak annotations in the form of scribbles. However, these approaches require large amounts of scribble annotations, and are only applied to the segmentation of regular organs, which are often unavailable for the disease species that fall in the long-tailed distribution. Motivated by the fact that the medical labels have anatomy distribution priors, we propose a scribble-supervised clustering-based framework, called MedCL, to learn the inherent anatomy distribution of medical labels. Our approach consists of two steps: i) Mix the features with intra- and inter-image mix operations, and ii) Perform feature clustering and regularize the anatomy distribution at both local and global levels. Combined with a small amount of weak supervision, the proposed MedCL is able to segment both regular organs and challenging irregular pathologies. We implement MedCL based on SAM and UNet backbones, and evaluate the performance on three open datasets of regular structure (MSCMRseg), multiple organs (BTCV) and irregular pathology (MyoPS). It is shown that even with less scribble supervision, MedCL substantially outperforms the conventional segmentation methods. Our code is available at https://github.com/BWGZK-keke/MedCL.

**Keywords:** Weakly supervised learning, Segmentation, Scribble, Data augmentation

## 1. Introduction

Manually labeling medical images is an arduous task that requires remarkable efforts from clinical experts. To alleviate this challenge, many recent approaches use sparsely annotated data for model training, termed weakly supervised learning (WSL) (Lin et al., 2016; Bai et al., 2018; Ji et al., 2019; Luo et al., 2022; Han et al., 2024). WSL leverages weak supervision such as scribbles, points, bounding boxes, and image-level labels (Tajbakhsh et al., 2020), by modeling shape priors (Kervadec et al., 2021) and developing novel loss functions (Kervadec et al., 2019). Existing WSL medical image segmentation methods focus mainly on scribbles, which are suitable to annotate nested structures (Can et al., 2018). These methods make use of large amounts of scribble annotations to compensate for the lack of fully supervised data. The existing scribble-supervised segmentation approaches can be divided into two categories. The first group of works aims to generate pseudo-labels that are then used for supervised training (Luo et al., 2022; Bai et al., 2018; Lin et al., 2016; Ji et al., 2019; Han et al., 2024). However, these models are susceptible to noise introduced by inaccurate segmentation results. The second line of approaches focuses on regularization techniques, to constrain the prediction with size priors (Zhang and Zhuang, 2022b) and the

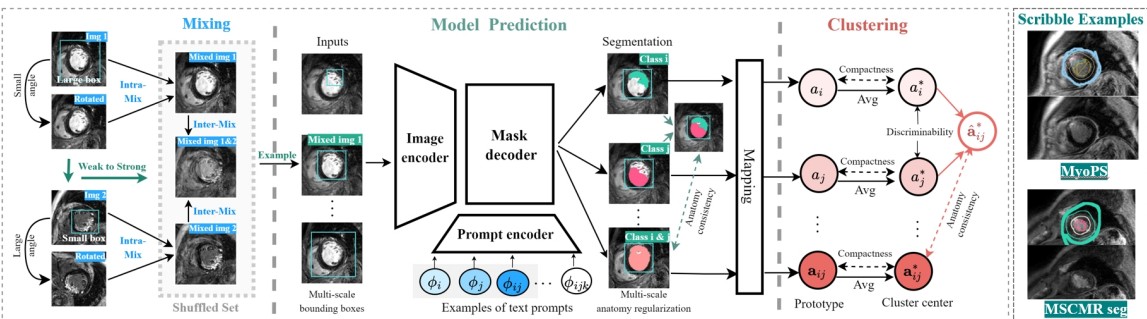

Figure 1: An overview of the proposed MedCL based on the SAM architecture.

consistency of the augmented versions (Zhang and Zhuang, 2022a; Zhang and Patel, 2024). These techniques work for regular organs, but sometimes fail in capturing the characteristics of irregular pathologies. Unlike existing methods, we propose to investigate the inherent anatomy distribution priors and take it as the principle to guide the image segmentation.

There exists a group of mix-based augmentation techniques (Zhang et al., 2018; DeVries and Taylor, 2017; Yun et al., 2019; Kim et al., 2020, 2021), termed as mixup. The mixup operation could lead to unrealistic results and change the shape of features significantly. To overcome this, several techniques (Zhang and Zhuang, 2022b,a; Zhang and Patel, 2024) have been proposed to leverage mix-invariant properties for regularization. However, these works treat mix-up as a regularization strategy and fail to increase the diversity of the original training samples. To tackle the above mentioned problems, we propose MedCL to learn anatomy distribution in an unsupervised manner. Existing unsupervised representation learning methods (Caron et al., 2018, 2019, 2020; Chen and He, 2021; Chen et al., 2020a; Wu et al., 2018; Chaitanya et al., 2020; Wu et al., 2024) are primarily designed for pre-training and often rely on large batch sizes to achieve optimal performance. In contrast, MedCL adopts a two-stage process of feature mixing and clustering, enabling models to be trained from scratch while effectively capturing the anatomical patterns inherent in medical semantics. Firstly, we propose a weak-to-strong mix strategies to thoroughly mix image features at both intra- and inter- image levels. Secondly, we perform the feature clustering and require the compactness within center, discriminability between centers, and the consistent anatomy distribution across all centers.

Our contributions are summarized as follows: 1) We propose a novel unsupervised clustering-based framework, *i.e.*, MedCL, to learn the anatomy distribution priors for medical image segmentation, with two implementations using SAM (Kirillov et al., 2023) and UNet (Ronneberger et al., 2015) backbones. 2) A feature-shuffling mechanism has been derived to generate a variety of image-prompt pairs for clustering. 3) We introduce the feature clustering approach to learn the inherent anatomy relationships among semantics. Specifically, we apply constraints to obtain compact, distinguishable, and regularly distributed feature clusters. 4) Evaluated on three open datasets,the proposed MedCL demonstrates promising performance significantly better than existing methods with fewer scribbles.

MEDCL

## 2. Method

As shown in Figure 1, MedCL is composed of two steps, including feature mixing and clustering. Firstly, we mix image features at intra- and inter- image levels. Then, the feature clustering is conducted with the online mapping and is regularized with anatomy properties. Finally, we apply MedCL to the medical image segmentation task, and combine it with the weak supervision, *i.e.*, scribble and image-level labels, to achieve better performance.

### 2.1. Mixing features

**Intra-mix:** We surmise that the image rotated with a small angle is a resemble of the artifact. Therefore we propose to mix the image $x$ of dimension $h \times w$ with its rotated version $R(x, \theta)$. We sample the intra-image mix ratio $\beta'$ from the beta distribution and obtain the mixed image $x'$ as $x' = \beta'x + (1 - \beta')R(x, \theta)$. Correspondingly, we define $y'$ as the segmentation of mixed image $x'$. We further introduce bounding boxes to enable the multi-scale mix while preserving the shape characteristics within the region of interest. We first randomly sample bounding boxes from the image, and train model to predict the segmentation results of the bounding boxes. Then, we perform mixup outside of the bounding boxes. Denoting the bounding box with the binary mask $I_b$, we modify the mix operation as $x' = I_bx + (1 - I_b)[\beta'x + (1 - \beta')R(x, \theta)]$.

To facilitate model training, we randomly sample text prompts for each class and generate their combinations. For $m$ classes, we begin by sampling text prompts $\phi_i$ for each class $\omega_i$, resulting in $\Phi_1 = \{\phi_i\}_{i=1}^m$, which instructs the model to generate segmentation for individual classes. Next, we sample class subsets $\Omega_k$ of size $k$ and derive the corresponding text prompt combinations $\Phi'_k = \{\phi_i \mid i \in \Omega_k\}$. For simplicity, we progressively increase $k$ from 2 to $m$, allowing $\Omega_k$ to gradually expand until it covers the entire class set. The resulting text prompt combinations are denoted as: $\mathbf{\Phi} = \{\Phi_1, \Phi'_2, \ldots, \Phi'_m\}$, These combinations instruct the model to generate segmentation results for multiple combined classes. Let $(x', I_b, \Phi)$ represent the mixed pairs, including intra-mixed image, the bounding box mask, and the text prompts. The corresponding ground truth segmentation is denoted by $\boldsymbol{y} = \{y_i\}_{i=1}^m \cup \{y'_k\}_{k=2}^m$, where $\boldsymbol{y}$ has a dimension of $h \times w \times (2m-1)$. Each component $y_i$ and $y'_k$ has a dimension of $h \times w \times 1$. Here, $y_i$ represents the segmentation probability map for class $\omega_i$, while $y'_k$ is the combined segmentation of class subset $\Omega_k$, defined as $y'_k = \bigcup_{i \in \Omega_k} y_i$. The union operation defines the total area covered by the segmentation labels.

**Inter-mix:** To perform the mix across images, we simply mix two images, fuse the bounding boxes, and interpolate the text tokens accordingly. Let $(x'_1, I_{b_1}, \mathbf{\Phi}_1)$ and $(x'_2, I_{b_2}, \mathbf{\Phi}_2)$ be the two training sample pairs, we derive the mixed samples $(x_{12}, I_{b_{12}}, \mathbf{\Phi}_{12})$ as $x_{12} = \beta x'_1 + (1-\beta)x'_2$, $I_{b_{12}} = I_{b_1} \cup I_{b_2}$, $\mathbf{\Phi}_{12} = \beta e(\mathbf{\Phi}_1) + (1-\beta)e(\mathbf{\Phi}_2)$, respectively. The text tokens $e(\mathbf{\Phi})$ are extracted with a prompt encoder $e(\cdot)$, and the inter-mix ratio $\beta$ is sampled from the beta distribution. The prediction of $x_{12}$ is denoted as $\hat{\boldsymbol{y}}_{12} = M(\hat{\boldsymbol{y}}'_1, \hat{\boldsymbol{y}}'_2) = \beta\boldsymbol{y}'_1 + (1-\beta)\hat{\boldsymbol{y}}'_2$, where the addition refers to pixel-wise addition of the probability maps. We thereby apply the mix consistency loss ($\mathcal{L}_{\text{mix}}$) to segmentation $\hat{\boldsymbol{y}}_{12} = f(x_{12}, I_{b_{12}}, \Phi_{12})$:

$$\mathcal{L}_{\text{mix}} = \text{sim}(\hat{\boldsymbol{y}}_{12}, M(\hat{\boldsymbol{y}}'_1, \hat{\boldsymbol{y}}'_2)), \tag{1}$$

where $\text{sim}(z_1, z_2) = -\frac{z_1 \cdot z_2}{\|z_1\|_2 \cdot \|z_2\|_2}$. we aim to minimize the negative cosine similarity between the mixed segmentation $M(\hat{\boldsymbol{y}}'_1, \hat{\boldsymbol{y}}'_2)$ and the segmentation of the mixed image $\hat{\boldsymbol{y}}_{12}$.

**Sampling:** We sample the augmented image multiple times to increase the number of features for clustering. Inspired by previous work (Caron et al., 2020), we first randomly crop regions of an image from a range of resolutions. Secondly, the intra- and inter- image mix are performed to obtain the mixed pairs of $(x', I_b, \Phi)$. Finally, we repeatedly sample training images from the mixed pairs to completely blend the features within the entire database, and achieve about 40 times amplification of training samples for each epoch.

## 2.2. Cluster features

**Online mapping:** For prediction $\hat{y}$ with dimensions $(2m-1) \times h \times w$, we aim to map it to a set of anatomical prototypes $\boldsymbol{a} = [a_1, \dots, a_d]$ of size $(2m-1) \times d$. By flattening the predictions, the multi-label probability map $\hat{y}$ is reshaped to dimension $(2m-1) \times n$, where $n = h \times w$. A mapping $P$ of size $d \times n$ is defined to maximize the similarity between the prediction $\hat{y}$ and the prototypes $\boldsymbol{a}$. The optimization objective is formulated as follows:

$$\max_{P \in \mathcal{R}^{d \times n}} \mathrm{Tr}(P^T \boldsymbol{a}^T \hat{y}) - w \sum_{i=1}^{d \times n} P_i \log P_i, \tag{2}$$

where the second term with weight $w$ is taken for regularization, aimed to control the smoothness of mapping $P$. The solution of Eq.(2) is denotes as $P^*$, which is derived as the normalized exponential matrix (Cuturi, 2013):

$$P^* = \mathrm{Diag}(U) \exp\left(\frac{\boldsymbol{a}^T \hat{y}}{w}\right) \mathrm{Diag}(V), \tag{3}$$

where $U \in \mathcal{R}^d$ and $V \in \mathcal{R}^n$ indicate the re-normalization vectors, which are efficiently determined using the Sinkhorn-Knopp algorithm (Cuturi, 2013; Caron et al., 2020). We optimize the algorithm on a per-batch basis; the corresponding pseudocode is provided in Appendix Section 7.

**Anatomy regularization:** We assume the anatomy prototype clusters should meet the following criteria: (1) Compactness: the density of prototype distribution within clusters. (2) Discriminability: Clear boundaries between clusters. (3) Anatomy consistency: Consistent distribution priors across all clusters. The cluster loss $\mathcal{L}_{\text{cluster}}$ is defined as:

$$\mathcal{L}_{\text{cluster}} = -\log\left[\frac{\sum_{b,i} \exp(\frac{1}{\tau}\mathrm{sim}(\boldsymbol{a}_i^b, \boldsymbol{a}_i^*))}{\sum_{b,i} \exp(\frac{1}{\tau}\mathrm{sim}(\boldsymbol{a}_i^b, \boldsymbol{a}_i^*)) + \sum_{i,j} \mathbb{1}_{i \neq j} \exp(\frac{1}{\tau}\mathrm{sim}(\boldsymbol{a}_i^*, \boldsymbol{a}_j^*))}\right], \tag{4}$$

where $\boldsymbol{a}_i^*$ and $\boldsymbol{a}_j^*$ $(i, j \in [1, m])$ denotes the cluster center of prototypes $\{\boldsymbol{a}_i^b\}_{b=1}^B$ for class $\omega_i$, which is calculated by $\boldsymbol{a}_i^* = \frac{1}{B}\sum_{b=1}^B \boldsymbol{a}_i^b$; $b$ refers to the index of samples within the batches of size $B$; $\tau$ is the temperature parameter controlling sharpness (Wu et al., 2018). Then, $\sum_{b,i} \exp(\frac{1}{\tau}\mathrm{sim}(\boldsymbol{a}_i^b, \boldsymbol{a}_i^*))$ controls the compactness and $\sum_{i,j} \mathbb{1}_{i \neq j} \exp(\frac{1}{\tau}\mathrm{sim}(\boldsymbol{a}_i^*, \boldsymbol{a}_j^*))$ represents the discriminability term.

For the third principle, *i.e.*, consistent anatomy distribution, we apply the consistency constraint to segmentation and prototypes at both global and local levels. By manipulating the text prompts, we construct the multi-scale regularization for model prediction. For the

class set $\Omega_k$ of size from 2 to $m$, we define the anatomy consistency loss $\mathcal{L}_{ac}$:

$$\mathcal{L}_{ac} = \sum_{j=m+1}^{2m-1} [\text{sim}(\hat{\boldsymbol{y}}_j, \sum_{i \in \Omega_k} \hat{\boldsymbol{y}}_i) + \text{sim}(\hat{\boldsymbol{a}}_j^*, \sum_{i \in \Omega_k} \hat{\boldsymbol{a}}_i^*)], \tag{5}$$

where $k = j - m + 1$, indicating that the number of categories within $\Omega_k$ increases along with $j$, thereby achieving the multi-scale constraint of distribution from local to global level. The first term applies the consistency of segmentation and the second term regularizes the anatomy distribution across prototype clusters.

**Weak supervision:** Although MedCL is conducted in a unsupervised setting, it can be easily combined with weak supervision forms of scribbles and image-level labels. For scribble annotations, we calculate the cross-entropy loss and dice loss for the annotated pixels, and thereby define the scribble-supervised loss as $\mathcal{L}_{\text{scribble}} = -\sum_{i=1}^{m} [\boldsymbol{y}_i \log(\hat{\boldsymbol{y}}_i) + 2\boldsymbol{y}_i\hat{\boldsymbol{y}}_i/(\boldsymbol{y}_i + \hat{\boldsymbol{y}}_i)]$ where $\boldsymbol{y}_i$ denotes the scribble annotations. For image-level labels, we exploit the given set of categories ($\Psi$) presented in the image, and require the sum of their probabilities equal to 1. The weakly-supervised loss of image-level labels is formulated accordingly as: $\mathcal{L}_{\text{category}} = -\log\left(\sum_{i \in \Psi} \boldsymbol{y}_i\right)$, which also minimizes the probability of non-exist classes. Then, the training objective $\mathcal{L}$ is derived as:

$$\mathcal{L} = \underbrace{\mathcal{L}_{\text{mix}} + \mathcal{L}_{\text{cluster}} + \mathcal{L}_{\text{ac}}}_{\text{unsupervised}} + \underbrace{\mathcal{L}_{\text{scribble}} + \mathcal{L}_{\text{category}}}_{\text{supervised}}. \tag{6}$$

## 3. Experiments

**Datasets: MSCMRseg** (Gao et al., 2023; Zhuang, 2018) dataset is released by the MIC-CAI'19 multi-sequence cardiac MR Segmentation challenge. It comprises of late gadolinium enhancement (LGE) MRI images obtained from 45 patients who underwent cardiomyopathy, The organizers provided annotations for the left ventricle (LV), myocardium (MYO), and the right ventricle (RV) in these images. Following (Yue et al., 2019), we randomly partition the images from the 45 patients into three sets: 25 training images, 5 validation images, and 15 test cases. We adopt the manual scribble annotations released by (Zhang and Zhuang, 2022a). **MyoPS** (Luo and Zhuang, 2022; Qiu et al., 2023) was released in the MICCAI'20 myocardial pathology segmentation challenge, which contains 45 paired multisequence CMR images of BSSFP, LGE and T2 CMR. MyoPS is a more challenging task compared to MSCMR structure segmentation, due to the heterogeneous representation of pathology across different patients. We use scribble annotations released by (Zhang and Patel, 2024; Zhang and Zhuang, 2023). Following Li *et al.* (Li et al., 2023), we split the dataset into 20 pairs for training, 5 for validation, and 20 for testing. **BTCV** (Landman et al., 2015) dataset contains 3D abdominal CT scans from 30 subjects, with annotations provided for 13 organs. Each scan comprises 80 to 225 slices at a resolution of $512\times512$ pixels. We employ complete annotations and identical data splits as those used in previous studies to ensure consistency and enable direct comparisons (Tang et al., 2022; Wu et al., 2024), with 24 images for training and 6 images for validation. Additionally, we include the results of a 5-fold cross-validation in the appendix (Sec.5) for reference. **Preprocessing:** For MSCMRseg and MyoPS, we extract a $256\times256$ central region for experiments with the

Table 1: Regular structure segmentation: Dice and HD comparison of MedCL on the MSCMRseg test set with 5 training scribbles.

| Methods | Backbone | Dice | | | | HD(mm) | | | |
|---|---|---|---|---|---|---|---|---|---|
| | | LV | MYO | RV | Avg | LV | MYO | RV | Avg |
| PCE | UNet | .261±.106 | .193±.095 | .018±.013 | .157±.132 | 259.43±14.19 | 240.58±13.41 | 254.20±12.66 | 251.40±15.39 |
| MixUp (Zhang et al., 2018) | UNet | .440±.102 | .310±.127 | .021±.013 | .257±.200 | 259.42±14.18 | 210.00±12.37 | 251.98±15.6 | 240.47±25.96 |
| Cutout (DeVries and Taylor, 2017) | UNet | .315±.103 | .307±.153 | .166±.110 | .263±.139 | 259.42±14.18 | 240.06±16.38 | 252.18±15.42 | 250.56±17.04 |
| CutMix (Yun et al., 2019) | UNet | .335±.119 | .282±.099 | .017±.013 | .211±.166 | 259.43±14.19 | 241.30±13.94 | 258.51±12.91 | 253.08±15.81 |
| Puzzle Mix (Kim et al., 2020) | UNet | .084±.029 | .351±.104 | .010±.008 | .148±.160 | 259.43±14.19 | 223.22±13.02 | 256.37±12.56 | 246.34±21.05 |
| Co-mixup (Kim et al., 2021) | UNet | .322±.169 | .221±.097 | .034±.010 | .192±.163 | 259.43±14.19 | 239.02±13.25 | 257.04±12.60 | 251.83±15.98 |
| CycleMix (Zhang and Zhuang, 2022a) | UNet | .517±.086 | .421±.108 | .007±.007 | .315±.237 | 213.20±35.65 | 151.36±55.12 | 260.56±12.66 | 208.37±58.88 |
| ShapePU (Zhang and Zhuang, 2022b) | UNet | .758±.191 | .567±.168 | .059±.026 | .461±.331 | 209.04±16.09 | 234.08±18.15 | 237.86±14.13 | 226.99±20.45 |
| WSL4 (Luo et al., 2022; Han et al., 2024) | UNet | .809±.079 | .653±.109 | .599±.261 | .687±.191 | 140.95±69.06 | 147.74±59.93 | 95.07±60.53 | 127.92±67.49 |
| ModelMix (Zhang and Patel, 2024) | UNet | .875±.077 | .754±.079 | .722±.201 | .784±.145 | 78.05±16.11 | 69.85±30.45 | 99.20±46.81 | 82.36±35.09 |
| **MedCL** | SAM | .882±.065 | .745±.065 | .789±.103 | .805±.065 | 7.50±4.54 | 10.85±6.56 | 26.47±12.48 | 14.94±5.23 |
| | UNet | **.904±.045** | **.787±.047** | **.804±.101** | **.832±.086** | 56.99±42.85 | 52.92±42.81 | 55.41±38.58 | 55.10±40.54 |
| FullSup-UNet | UNet | .775±.158 | .604±.147 | .572±.207 | .651±.191 | 23.50±21.79 | 34.03±19.25 | 81.29±11.29 | 46.27±30.91 |
| FullSup-nnUNet | nnUNet | .885±.085 | .757±.147 | .757±.201 | .799±.160 | 21.48±29.68 | 13.5±12.99 | 18.27±12.51 | 17.75±19.87 |

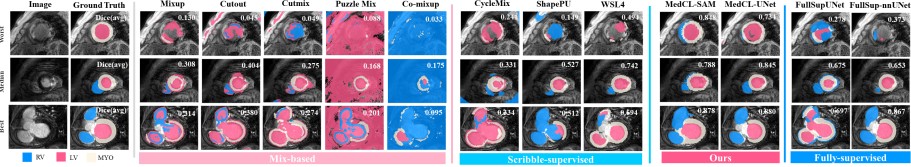

Figure 2: The visualization of regular organ segmentation from the MSCMRseg dataset.

UNet (Ronneberger et al., 2015) backbone, while for BTCV, we use nnU-Net (Isensee et al., 2021). For SAM-based models, the region is resized to 1024×1024, and slice intensities are normalized to [0,1]. Models are trained with a learning rate of $1e^{-4}$ on eight NVIDIA RTX A5000 GPUs. **Implementation:** The two versions of MedCL, based on SAM (Kirillov et al., 2023) and UNet (Ronneberger et al., 2015) architecture, are termed as MedCL-SAM and MedCL-UNet, respectively. For MedCL-SAM, we leverage the pretrained encoder of MedSAM (Ma and Wang, 2023) and fine-tune its decoder. The text prompts are encoded using the pre-trained CLIP (Radford et al., 2021). For MedCL-UNet, we disable the input of bounding box and text prompts, and train the model from scratch. **Evaluation Metrics**: Following the practice of medical image segmentation, we report the Dice score and the Hausdorff distance (HD) for each foreground class of MSCMRseg and MyoPS segmentation tasks separately.

### 3.1. Results

**Regular structure segmentation:** Table 1 presents the Dice and HD results for cardiac ventricle segmentation using *twelve* methods on the MSCMRseg dataset. Results for ModelMix (Zhang and Patel, 2024) are borrowed from the original publication, while other methods are implemented by us. Note that ModelMix trains models on complementary datasets, while our approach focuses on a single task without requiring additional data sources. Both MedCL-SAM and MedSAM-UNet outperform the compared approaches by large margins in terms of Dice and HD metrics. This is further affirmed by the qualitative results in Figure 2, which visualizes the worst and median cases selected based on the average Dice score. The poor performance of PuzzleMix and Co-Mixup is caused by their patch transportation strategy, a limitation noted in previous work (Zhang and Zhuang, 2022a).

Table 2: Multi-organ segmentation on BTCV multi-organ segmentation (Dice).

| Type | Method | Spl | RKid | LKid | Gall | Eso | Liv | Sto | Aor | IVC | Veins | Pan | RAG | LAG | AVG |
|---|---|---|---|---|---|---|---|---|---|---|---|---|---|---|---|
| w/ general pretraining | MAE3D (Chen et al., 2023; He et al., 2022) | 93.98 | 94.37 | 94.18 | 69.86 | 74.65 | 96.66 | 80.40 | 90.30 | 83.10 | 72.65 | 77.11 | 71.34 | 60.54 | 81.33 |
| | SimCLR (Chen et al., 2020b) | 92.79 | 93.52 | 93.36 | 60.24 | 60.64 | 95.90 | 79.92 | 85.56 | 80.58 | 63.47 | 67.77 | 55.99 | 50.45 | 75.14 |
| | SimMIM (Xie et al., 2022) | 95.56 | 95.56 | 95.08 | 63.56 | 53.52 | 98.98 | 90.42 | 92.71 | 85.82 | 58.63 | 71.16 | 60.55 | 47.73 | 78.88 |
| | MoCo v3 (He et al., 2020; Chen et al., 2021b) | 94.92 | 93.85 | 92.42 | 65.28 | 62.77 | 96.89 | 78.64 | 88.66 | 82.21 | 71.15 | 75.09 | 66.48 | 58.81 | 79.54 |
| | Jigsaw (Chen et al., 2021a) | 94.62 | 93.45 | 93.23 | 75.63 | 73.23 | 95.03 | 85.61 | 90.65 | 83.58 | 71.71 | 79.57 | 65.68 | 58.05 | 81.35 |
| | PositionLabel (Zhang and Gong, 2023) | 94.35 | 93.15 | 93.21 | 75.39 | 72.34 | 95.55 | 87.94 | 90.34 | 84.41 | 71.18 | 79.02 | 65.11 | 60.12 | 81.09 |
| w/ medical pretraining | MG (Zhou et al., 2021) | 91.99 | 93.52 | 91.81 | 65.11 | 76.14 | 95.98 | 86.88 | 89.65 | 83.59 | 71.79 | 81.50 | 67.97 | 63.18 | 81.45 |
| | ROT (Taleb et al., 2020) | 91.75 | 93.18 | 91.62 | 65.09 | 76.55 | 95.85 | 86.16 | 89.74 | 83.03 | 71.73 | 81.51 | 67.07 | 62.90 | 81.25 |
| | Vicreg (Bardes et al., 2022) | 92.03 | 92.50 | 91.62 | 75.24 | 74.96 | 96.07 | 85.50 | 89.43 | 83.08 | 74.74 | 78.35 | 71.14 | 63.44 | 81.81 |
| | Rubik++ (Tao et al., 2020) | 96.21 | 91.36 | 92.68 | 75.22 | 75.52 | 97.44 | 85.94 | 89.76 | 82.96 | 74.47 | 79.25 | 71.13 | 62.10 | 82.39 |
| | PCRL (Zhou et al., 2023) | 95.30 | 91.43 | 89.62 | 76.15 | 72.58 | 95.88 | 86.15 | 89.08 | 83.42 | 75.13 | 80.17 | 67.50 | 62.73 | 81.85 |
| | Swin-UNETR (Tang et al., 2022) | 95.21 | 92.03 | 92.22 | 74.27 | 73.39 | 96.32 | 84.62 | 90.78 | 83.03 | 75.51 | 79.87 | 68.99 | 61.59 | 82.11 |
| | SwinMIM (Wang et al., 2023) | 95.44 | 92.43 | 94.37 | 75.29 | 73.06 | 96.44 | 84.20 | 90.76 | 83.10 | 70.91 | 79.78 | 70.11 | 62.44 | 82.07 |
| | GL-MAE (Zhuang et al., 2023) | 95.21 | 91.22 | 92.37 | 76.19 | 73.66 | 96.09 | 86.23 | 89.80 | 81.65 | 75.71 | 79.68 | 70.36 | 60.98 | 81.92 |
| | GVSL (He et al., 2023) | 95.27 | 91.22 | 92.37 | 74.92 | 74.20 | 96.64 | 86.02 | 90.48 | 82.14 | 72.42 | 78.67 | 67.44 | 62.73 | 81.93 |
| | VoCo (Wu et al., 2024) | 95.73 | 96.53 | 94.48 | 76.02 | 76.50 | 97.41 | 78.43 | 91.21 | 86.12 | 78.19 | 80.88 | 71.47 | 67.88 | 83.85 |
| from scratch | UNETR (Hatamizadeh et al., 2022) | 93.02 | 94.13 | 94.12 | 66.99 | 70.87 | 96.11 | 77.27 | 89.22 | 82.10 | 70.16 | 76.65 | 65.32 | 59.21 | 79.82 |
| | Swin-UNETR (Hatamizadeh et al., 2021) | 94.06 | 93.54 | 93.80 | 65.51 | 74.60 | 97.09 | 75.94 | 91.80 | 82.36 | 73.63 | 75.19 | 68.00 | 61.11 | 80.53 |
| | MedCL-nnUNet | 96.77 | 95.29 | 95.32 | 62.95 | 76.71 | 97.30 | 90.43 | 90.48 | 88.91 | 79.47 | 86.53 | 74.52 | 73.05 | 85.21 |

Table 3: Irregular pathology segmentation: comparison on MyoPS test set using 5 scribbles.

| Methods | Backbone | Dice | | | HD | | |
|---|---|---|---|---|---|---|---|
| | | Scar | Edema | Avg | Scar | Edema | Avg |
| PCE | UNet | .242±.170 | .122±.077 | .182±.144 | 76.22±37.24 | 124.89±21.27 | 100.55±38.77 |
| nnPU (Kiryo et al., 2017) | UNet | .290±.166 | .236±.078 | .263±.131 | 126.51±35.27 | 125.05±20.69 | 125.78±28.55 |
| CVIR (Garg et al., 2021) | UNet | .288±.191 | .085±.034 | .186±.170 | 45.01±18.44 | 125.27±20.83 | 85.14±45.04 |
| ModelMix (Zhang and Patel, 2024) | UNet | .348±.189 | .531±.106 | .440±.177 | - | - | - |
| MedCL | SAM | .467±.222 | .505±.113 | .486±.155 | 28.06±13.01 | 31.88±10.57 | 29.97±6.76 |
| | UNet | .458±.229 | .536±.152 | .497±.196 | 47.84±18.40 | 42.91±17.64 | 45.37±17.97 |
| FullSup-UNet | UNet | .423±.253 | .445±.149 | .434±.205 | 117.61±35.08 | 119.13±22.7 | 118.37±29.17 |
| FullSup-nnUNet | nnUNet | .496±.252 | .563±.141 | .529±.204 | 43.86±37.27 | 45.14±33.86 | 44.50±35.15 |

Figure 3: Qualitative results of typical pathologies from MyoPS dataset.

**Multi-organ segmentation:** Table 4 presents the multi-organ segmentation results on the BTCV dataset. The results of the compared methods are sourced from VOCO (Wu et al., 2024), which was pre-trained on 1.6K CT scans (including BTCV) and further fine-tuned using the BTCV dataset. Remarkably, even without pre-training, our model trained from scratch outperforms VOCO by an average Dice of 1.72%. This highlights the effectiveness of our proposed method in capturing the anatomical distribution of multiple organs.

**Irregular pathology segmentation:** We further evaluate MedCL to the challenging task of myocardial pathology segmentation (MyoPS) with heterogeneous shape features. We compare the proposed MedCL with scribble-supervised (nnPU (Kiryo et al., 2017), CVIR (Garg et al., 2021), ModelMix (Zhang and Patel, 2024)) and fully-supervised (Fullsup-UNet, Fullsup-nnUNet) benchmarks. We borrow the results of ModelMix from the original paper, without incorporating any additional data sources. Table 3 summarizes the results. One can find that the advantages of the MedCL are demonstrated evidently in such a difficult task, achiving comparable performance with fully supervised benchmarks with SAM and UNet backbones, respectively. Figure 3 presents three typical cases.

Table 4: Results on BTCV. The best results are bolded. 'From Scratch' denotes the supervised baseline without self-supervised pretraining.

| Type | Method | Spl | RKid | LKid | Gall | Eso | Liv | Sto | Aor | IVC | Veins | Pan | RAG | LAG | AVG |
|---|---|---|---|---|---|---|---|---|---|---|---|---|---|---|---|
| w/ general pretraining | MAE3D (Chen et al., 2023; He et al., 2022) | 93.98 | 94.37 | 94.18 | 69.86 | 74.65 | 96.66 | 80.40 | 90.30 | 83.10 | 72.65 | 77.11 | 71.34 | 60.54 | 81.33 |
| | SimCLR (Chen et al., 2020b) | 92.79 | 93.52 | 93.36 | 60.24 | 60.64 | 95.90 | 79.92 | 85.56 | 80.58 | 63.47 | 67.77 | 55.99 | 50.45 | 75.14 |
| | SimMIM (Xie et al., 2022) | 95.56 | 95.56 | 95.08 | 63.56 | 53.52 | 98.98 | 90.42 | 92.71 | 85.82 | 58.63 | 71.16 | 60.55 | 47.73 | 78.88 |
| | MoCo v3 (He et al., 2020; Chen et al., 2021b) | 94.92 | 93.85 | 92.42 | 65.28 | 62.77 | 96.89 | 78.64 | 88.66 | 82.21 | 71.15 | 75.09 | 66.48 | 58.81 | 79.54 |
| | Jigsaw (Chen et al., 2021a) | 94.62 | 93.45 | 93.23 | 75.63 | 73.23 | 95.03 | 85.61 | 90.65 | 83.58 | 71.71 | 79.57 | 65.68 | 58.05 | 81.35 |
| | PositionLabel (Zhang and Gong, 2023) | 94.35 | 93.15 | 93.21 | 75.39 | 72.34 | 95.55 | 87.94 | 90.34 | 84.41 | 71.18 | 79.02 | 65.11 | 60.12 | 81.09 |
| w/ medical pretraining | MG (Zhou et al., 2021) | 91.99 | 93.52 | 91.81 | 65.11 | 76.14 | 95.98 | 86.88 | 89.65 | 83.59 | 71.79 | 81.50 | 67.97 | 63.18 | 81.45 |
| | ROT (Taleb et al., 2020) | 91.75 | 93.18 | 91.62 | 65.09 | 76.55 | 95.85 | 86.16 | 89.74 | 83.03 | 71.73 | 81.51 | 67.07 | 62.90 | 81.25 |
| | Vicreg (Bardes et al., 2022) | 92.03 | 92.50 | 91.62 | 75.24 | 74.96 | 96.07 | 85.50 | 89.43 | 83.08 | 74.74 | 78.35 | 71.14 | 63.44 | 81.81 |
| | Rubik++ (Tao et al., 2020) | 96.21 | 91.36 | 92.68 | 75.22 | 75.52 | 97.44 | 85.94 | 89.76 | 82.96 | 74.47 | 79.25 | 71.13 | 62.10 | 82.39 |
| | PCRL (Zhou et al., 2023) | 95.30 | 91.43 | 89.62 | 76.15 | 72.58 | 95.88 | 86.15 | 89.08 | 83.42 | 75.13 | 80.17 | 67.50 | 62.73 | 81.85 |
| | Swin-UNETR (Tang et al., 2022) | 95.21 | 92.03 | 92.22 | 74.27 | 73.39 | 96.32 | 84.62 | 90.78 | 83.03 | 75.51 | 79.87 | 68.99 | 61.59 | 82.11 |
| | SwinMIM (Wang et al., 2023) | 95.44 | 92.43 | 94.37 | 75.29 | 73.06 | 96.44 | 84.20 | 90.76 | 83.10 | 70.91 | 79.78 | 70.11 | 62.44 | 82.07 |
| | GL-MAE (Zhuang et al., 2023) | 95.21 | 91.22 | 92.37 | 76.19 | 73.66 | 96.09 | 86.23 | 89.80 | 81.65 | 75.71 | 79.68 | 70.36 | 60.98 | 81.92 |
| | GVSL† (He et al., 2023) | 95.27 | 91.22 | 92.37 | 74.92 | 74.20 | 96.64 | 86.02 | 90.48 | 82.14 | 72.42 | 78.67 | 67.44 | 62.73 | 81.93 |
| | VoCo (Wu et al., 2024) | 95.73 | 96.53 | 94.48 | 76.02 | 76.50 | 97.41 | 78.43 | 91.21 | 86.12 | 78.19 | 80.88 | 71.47 | 67.88 | 83.85 |
| from scratch | UNETR (Hatamizadeh et al., 2022) | 93.02 | 94.13 | 94.12 | 66.99 | 70.87 | 96.11 | 77.27 | 89.22 | 82.10 | 70.16 | 76.65 | 65.32 | 59.21 | 79.82 |
| | Swin-UNETR (Hatamizadeh et al., 2021) | 94.06 | 93.54 | 93.80 | 65.51 | 74.60 | 97.09 | 75.94 | 91.80 | 82.36 | 73.63 | 75.19 | 68.00 | 61.11 | 80.53 |
| | ours | 95.69 | 95.04 | 96.60 | 84.24 | 78.65 | 97.57 | 90.60 | 93.78 | 87.22 | 75.87 | 80.99 | 72.64 | 63.47 | 85.57 |

**Supervision sensitivity:** By varying the number of scribble annotations, we validate the supervision sensitivity of MedCL and compare it to FullSup-nnUNet and scribble-supervised model on MSCMRseg and MyoPS, *i.e.*, WSL4, and nnPU. As shown in Figure 4 (a) and (b), our MedCL surpasses the scribble-supervised benchmarks by large margins on all experiments. Interestingly, one can observe that when the number of annotations is particularly small (less than 5), scribble-supervised MedCL achieves comparable or even slightly better performance than FullSup-nnUNet. This indicates the superiority of MedCL in the situation of extremely weak supervision.

**Comparison to SAM-based methods:** By changing the shift of the bounding box prompts, we evaluate the robustness of MedCL-SAM against noisy bounding boxes on MSCMRseg and MyoPS datasets. As shown in Figure 4 (c) and (d), the performance of SAM and MedSAM clearly decreases as the bounding box shift increases. In contrast, MedCL-SAM is robust to bounding box shifts. Thanks to feature shuffling and anatomy-guided clustering, MedCL achieves the stable performance against noisy prompts.

**Ablations:** We validate the effectiveness of MedCL components with SAM and UNet backbones. The models are trained with 5 scribbles and evaluated on the validation set. We verify the key components of MedCL, including feature mixing ($\text{Mix}/\mathcal{L}_{\text{mix}}$), cluster loss ($\mathcal{L}_{\text{cluster}}$, Eq. 4), and anatomy consistency loss ($\mathcal{L}\text{ac}$, Eq. 5). Details are summarized in Table 5. Incorporating feature shuffling, Model #2 shows substantial improvement over Model #1, with an average Dice increase of 17.1% (0.521 vs. 0.350) for SAM and 34.1% (0.563 vs. 0.222) for UNet, highlighting the benefits of our mix augmentation. Model #3, enhanced with cluster loss ($\mathcal{L}_{\text{cluster}}$), further improves Dice by 20.8% (0.558 vs. 0.350) for UNet and 7.5% (0.638 vs. 0.563) for SAM. Finally, with the addition of anatomy consistency loss ($\mathcal{L}_{\text{ac}}$), our MedCL model achieves the best performance for SAM and UNet, respectively. We provide the ablations of text prompts (Sec.3), batch size (Sec.2), and investigate the effectiveness of feature shuffling components (Sec.4), category label (Sec.1) in Appendix.

# 4. Conclusion

In this work, we have presented MedCL, a novel framework to learn anatomy distribution for medical image segmentation, with two implementations based on SAM and UNet ar-

Table 5: Ablation on MSCMRseg validation (* p ≤ 0.05, Wilcoxon test).

| Methods | Mix/$\mathcal{L}_{\mathrm{mix}}$ | $\mathcal{L}_{\mathrm{cluster}}$ | $\mathcal{L}_{\mathrm{ac}}$ | MedCL-SAM (w/ text prompt) | | | | MedCL-UNet (w/o text prompt) | | | |
|---|---|---|---|---|---|---|---|---|---|---|---|
| | | | | LV | MYO | RV | Avg | LV | MYO | RV | Avg |
| #1 | × | × | × | .567±.315 | .317±.254 | .166±.112 | .350±.165 | .139±.131 | .275±.225 | .251±.194 | .222±.184 |
| #2 | ✓ | × | × | .806±.147* | .724±.060* | .032±.029 | .521±.070* | .762±.198* | .443±.119* | .484±.442* | .563±.304* |
| #3 | × | ✓ | × | .885±.051* | .606±.066 | .183±.300* | .558±.130 | .654±.202 | .581±.173* | .678±.304* | .638±.220* |
| #4 | ✓ | ✓ | × | .882±.052 | .718±.067* | .520±.286* | .707±.110* | .899±.087* | .693±.219* | .641±.379 | .744±.265* |
| MedCL | ✓ | ✓ | ✓ | .894±.067 | .784±.065* | .821±.122* | .833±.083* | .914±.059 | .779±.076* | .791±.198* | .828±.133* |

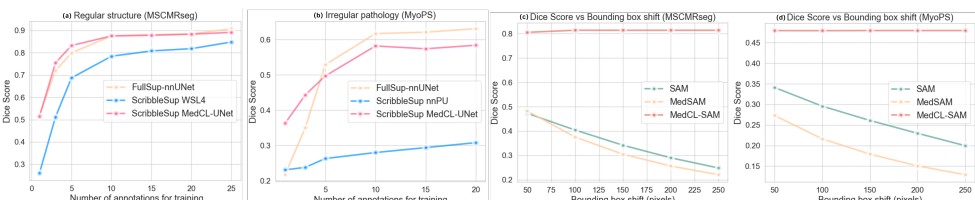

Figure 4: The impact of supervision amount (a,b) and bounding box shift (c,d).

chitectures. MedCL exploits supervision via feature mixing, and effectively learns anatomy priors with regularizations of compactness, discriminability, and distribution consistency. Evaluated on three challenging segmentation tasks, MedCL demonstrates state-of-the-art performance with robust and general applicability.

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

We evaluate the effect of loss calculations for category labels (Sec.1), batch size influence (Sec.2), text prompt types (Sec.3), and components of feature shuffling (Sec.4). Cross-validation results on BTCV are detailed in Sec.5, and typical scribbles from MSCMRseg and MyoPS are shown in Sec.6.

## .1. Category labels

Table I: The effect of label categories on MyoPS dataset.

| Method | $\mathcal{L}_{\text{sup-category}}$ | 1 scribble | | | 3 scribbles | | | 5 scribbles | | | 10 scribbles | | |
|---|---|---|---|---|---|---|---|---|---|---|---|---|---|
| | | Scar | Edema | AVG | Scar | Edema | AVG | Scar | Edema | AVG | Scar | Edema | AVG |
| nnPU | - | .230 | .192 | .231 | .397 | .080 | .238 | .290 | .236 | .263 | .477 | .084 | .280 |
| MedCL-UNet* | × | .302 | .262 | .282 | .402 | .426 | .414 | .449 | .522 | .486 | .470 | .527 | .498 |
| MedCL-UNet | ✓ | .218 | .508 | .363 | .391 | .495 | .443 | .458 | .536 | .497 | .517 | .647 | .582 |
| FullSup-nnUNet | - | .121 | .314 | .218 | .230 | .460 | .350 | .496 | .563 | .529 | .590 | .643 | .617 |

We compare the detailed performance of MedCL-UNet* (without $\mathcal{L}_{\text{sup-category}}$) and compared methods using MyoPS dataset, including MedCL-UNet, nnPU, and FullSup-nnUNet. We vary the training scribbles from 1 to 10, and summarize the detailed results in Table I. One can observe that even without $\mathcal{L}_{\text{sup-category}}$, MedCL-UNet* still evidently surpasses scribble-supervised method nnPU, demonstrating the effectiveness of proposed feature shuffling and clustering strategies.

## .2. Batch size

Table II: The effect of batch sizes on MSCMR dataset.

| Batch Size | LV | MYO | RV | Avg |
|---|---|---|---|---|
| 16 | .890±.076 | .773±.058 | .778±.087 | .814±.091 |
| 32 | .899±.048 | .784±.047 | .768±.163 | .817±.116 |
| 64 | .907±.045 | .787±.047 | .804±.047 | .832±.086 |

Table II reports the performance of MedCL-UNet with a batch size ranges from 16 to 64. All models are trained with 5 scribbles using MSCMRseg dataset. Our MedCL-UNet works consistently well over the range of batch size, with a slight drop of 1.8% (0.814 vs 0.832) for batch size 16 or 1.5% (0.817 vs 0.832) for batch size 32. This demonstrates that our MedCL performs robustly even on small batch sizes.

## .3. Types of text prompts

Table IV compares MedCL-SAM to MedSAM and SAM with determinstic and ambiguous text prompts. The determinstic prompt refers to noun, *i.e.*, "Myocardium", and "Left Ventrical". The ambiguous prompts refers to the sentence description, such as "Myocardium typically appears dark or black in LGE images, and have circular shape. SAM and MedSAM are the latest segmentation benchmarks pre-trained with large-scale natural and medical image datasets, respectively, while MedCL-SAM is initialized with the weights of MedCL and finetuned with 5 scribbles. One can find that our MedCL-SAM achieves promising

Table V: Multi-organ segmentation on BTCV (Dice): "From Scratch" denotes the supervised baseline without self-supervised pretraining.

| Type | Method | Spl | RKid | LKid | Gall | Eso | Liv | Sto | Aor | IVC | Veins | Pan | RAG | LAG | AVG |
|---|---|---|---|---|---|---|---|---|---|---|---|---|---|---|---|
| w/ general pretraining | MAE3D (Chen et al., 2023; He et al., 2022) | 93.98 | 94.37 | 94.18 | 69.86 | 74.65 | 96.66 | 80.40 | 90.30 | 83.10 | 72.65 | 77.11 | 71.34 | 60.54 | 81.33 |
| | SimCLR (Chen et al., 2020b) | 92.79 | 93.52 | 93.36 | 60.24 | 60.64 | 95.90 | 79.92 | 85.56 | 80.58 | 63.47 | 67.77 | 55.99 | 50.45 | 75.14 |
| | SimMIM (Xie et al., 2022) | 95.56 | 95.56 | 95.08 | 63.56 | 53.52 | 98.98 | 90.42 | 92.71 | 85.82 | 58.63 | 71.16 | 60.55 | 47.73 | 78.88 |
| | MoCo v3 (He et al., 2020; Chen et al., 2021b) | 94.92 | 93.85 | 92.42 | 65.28 | 62.77 | 96.89 | 78.64 | 88.66 | 82.21 | 71.15 | 75.09 | 66.48 | 58.81 | 79.54 |
| | Jigsaw (Chen et al., 2021a) | 94.62 | 93.45 | 93.23 | 75.63 | 73.23 | 95.03 | 85.61 | 90.65 | 83.58 | 71.71 | 79.57 | 65.68 | 58.05 | 81.35 |
| | PositionLabel (Zhang and Gong, 2023) | 94.35 | 93.15 | 93.21 | 75.39 | 72.34 | 95.55 | 87.94 | 90.34 | 84.41 | 71.18 | 79.02 | 65.11 | 60.12 | 81.09 |
| w/ medical pretraining | MG (Zhou et al., 2021) | 91.99 | 93.52 | 91.81 | 65.11 | 76.14 | 95.98 | 86.88 | 89.65 | 83.59 | 71.79 | 81.50 | 67.97 | 63.18 | 81.45 |
| | ROT (Taleb et al., 2020) | 91.75 | 93.18 | 91.62 | 65.09 | 76.55 | 95.85 | 86.16 | 89.74 | 83.03 | 71.73 | 81.51 | 67.07 | 62.90 | 81.25 |
| | Vicreg (Bardes et al., 2022) | 92.03 | 92.50 | 91.62 | 75.24 | 74.96 | 96.07 | 85.50 | 89.43 | 83.08 | 74.74 | 78.35 | 71.14 | 63.44 | 81.81 |
| | Rubik++ (Tao et al., 2020) | 96.21 | 91.36 | 92.68 | 75.22 | 75.52 | 97.44 | 85.94 | 89.76 | 82.96 | 74.47 | 79.25 | 71.13 | 62.10 | 82.39 |
| | PCRL (Zhou et al., 2023) | 95.30 | 91.43 | 89.62 | 76.15 | 72.58 | 95.88 | 86.15 | 89.08 | 83.42 | 75.13 | 80.17 | 67.50 | 62.73 | 81.85 |
| | Swin-UNETR (Tang et al., 2022) | 95.21 | 92.03 | 92.22 | 74.27 | 73.39 | 96.32 | 84.62 | 90.78 | 83.03 | 75.51 | 79.87 | 68.99 | 61.59 | 82.11 |
| | SwinMIM (Wang et al., 2023) | 95.44 | 92.43 | 94.37 | 75.29 | 73.06 | 96.44 | 84.20 | 90.76 | 83.10 | 70.91 | 79.78 | 70.11 | 62.44 | 82.07 |
| | GL-MAE (Zhuang et al., 2023) | 95.21 | 91.22 | 92.37 | 76.19 | 73.39 | 96.09 | 86.23 | 89.80 | 81.65 | 75.71 | 79.68 | 70.36 | 60.98 | 81.92 |
| | GVSL (He et al., 2023) | 95.27 | 91.22 | 92.37 | 74.92 | 74.20 | 96.64 | 86.02 | 90.48 | 82.14 | 72.42 | 78.67 | 67.44 | 62.73 | 81.93 |
| | VoCo (Wu et al., 2024) | 95.73 | 96.53 | 94.48 | 76.02 | 76.50 | 97.41 | 78.43 | 91.21 | 86.12 | 78.19 | 80.88 | 71.47 | 67.88 | 83.85 |
| from scratch | UNETR (Hatamizadeh et al., 2022) | 93.02 | 94.13 | 94.12 | 66.99 | 70.87 | 96.11 | 77.27 | 89.22 | 82.10 | 70.16 | 76.65 | 65.32 | 59.21 | 79.82 |
| | Swin-UNETR (Hatamizadeh et al., 2021) | 94.06 | 93.54 | 93.80 | 65.51 | 74.60 | 97.09 | 75.94 | 91.80 | 82.36 | 73.63 | 75.19 | 68.00 | 61.11 | 80.53 |
| | **MedCL**(Same split with compared methods) | **96.77** | 95.29 | 95.32 | 62.95 | 76.71 | 97.30 | 90.43 | 90.48 | **88.91** | **79.47** | **86.53** | **74.52** | **73.05** | 85.21 |
| | **MedCL**(Cross-validation) | 95.69 | 95.04 | **96.60** | **84.24** | **78.65** | 97.57 | 90.60 | 93.78 | 87.22 | 75.87 | 80.99 | 72.64 | 63.47 | **85.57** |
| MedCL (Cross-validation) | Fold1 | 95.66 | 95.72 | 96.70 | 74.39 | 81.18 | 96.77 | 73.93 | 92.33 | 93.64 | 84.09 | 74.20 | 73.46 | 64.62 | 84.36 |
| | Fold2 | 97.35 | 92.33 | 96.66 | 72.87 | 82.78 | 97.87 | 94.83 | 93.92 | 87.51 | 88.25 | 83.12 | 84.50 | 80.58 | 88.66 |
| | Fold3 | 92.93 | 95.44 | 96.31 | 89.47 | 75.30 | 97.67 | 94.07 | 94.21 | 75.06 | 65.68 | 78.01 | 72.69 | 27.73 | 81.12 |
| | Fold4 | 96.81 | 96.11 | 96.66 | 93.89 | 77.28 | 97.86 | 95.17 | 94.52 | 89.72 | 56.65 | 83.23 | 61.95 | 87.54 | 86.72 |
| | Fold5 | 95.70 | 95.60 | 96.68 | 90.57 | 76.72 | 97.66 | 95.00 | 93.91 | 90.15 | 84.68 | 86.38 | 70.60 | 56.90 | 86.97 |

Table III: The influence of text prompt types on MSCMRseg dataset.

| Methods | Text prompt | LV | MYO | RV | Avg |
|---|---|---|---|---|---|
| SAM | Deterministic | .037±.013 | .042±.011 | .039±.011 | .040±.010 |
| MedSAM | Deterministic | .041±.034 | .024±.006 | .026±.009 | .030±.012 |
| MedCL-SAM | Deterministic | **.879±.078** | **.744±.066** | **.793±.087** | **.805±.062** |
| SAM | Ambiguous | .040±.010 | .037±.013 | .042±.011 | .039±.011 |
| MedSAM | Ambiguous | .023±.007 | .026±.006 | .024±.008 | .024±.005 |
| MedCL-SAM | Ambiguous | **.633±.206** | **.563±.099** | **.399±.159** | **.532±.116** |

Table IV: Examples of text prompt on MSCMRseg dataset.

| Noun (Deterministic) | Description (Ambiguous) |
|---|---|
| RV (Right Ventricle) | Right ventricle has complex shape, triangular from the frontal aspect and crescentic from the apex. |
| Myo (Myocardium) | Myocardium typically appears dark or black in LGE images, and has a circular shape. |
| LV (Left Ventricle) | Left ventricle is typically observed as a roughly elliptical or oblong structure. |

results with various text prompts, although the performance decreases on ambiguous descriptions of long sentence. By contrast, SAM and MedSAM fail to tackle text prompts for these tasks, indicating the necessity of our proposed anatomy prior guided fine-tuning.

We provide the list of text prompts in the table Table IV. Using the MSCMR dataset as an example, there are three classes: RV, Myo, and LV. We utilize two groups of text prompts: the noun group and the sentence description group.

We sample class combinations following the pattern below. As described in the manuscript, we first sample text prompts for each class. Taking MSCMRseg as an example, there are three foreground classes: RV, Myo, and LV. The initial sampled prompt is therefore [RV, Myo, LV]. Next, we sample combinations of these classes, with the number of classes in each combination ranging from 2 to m (where m is the total number of foreground classes). For MSCMRseg, the possible combinations are: [RV and LV, RV and Myo, Myo and LV, RV and Myo and LV]. For example, when sampling two-class combinations, we might select [RV

Table V: Ablation study of intra-mix components on MSCMRseg.

| Methods | Intra-Mix | | Inter-Mix | Dice | | | |
| --- | --- | --- | --- | --- | --- | --- | --- |
| | Bounding box | Rotation | | LV | MYO | RV | Avg |
| #1 | × | × | × | .139 | .275 | .251 | .222 |
| #2 | ✓ | × | × | .700 | .431 | .296 | .476 |
| #3 | ✓ | ✓ | × | .661 | **.527** | .325 | .505 |
| #4 | × | × | ✓ | .700 | .416 | **.501** | .539 |
| #5 | ✓ | ✓ | ✓ | **.762** | .443 | .484 | **.563** |

and LV]. For three-class combinations, we get [RV and Myo and LV]. The final sampled set, in this case, could be [RV, Myo, LV, RV and LV, RV and Myo and LV], which has a dimension of 5. In general, for foreground classes, the total number of possible sampled prompts is (e.g., 2×3-1=5). This progressive sampling strategy allows us to gradually expand the set of class combinations until it covers all possible subsets of the class set.

Additionally, we apply data augmentation during the sampling process. For example, the conjunction "and" can be replaced with synonyms (e.g., "with", "along with"), and class names such as RV can be substituted with equivalent terms like Right Ventricle.

### .4. Components of feature mixing

For intra-mix, the bounding boxes control the range, while the rotated angles determine the intensity. The two operations work in an complementary way. Following the settings of ablations in Table 6, we provide the results of ablations in Table V.

### .5. Cross-validation on BTCV dataset

For a fair comparison, we report results using the same dataset split provided by VoCo (Wu et al., 2024), consistent with prior studies (Chen et al., 2023; Zhou et al., 2021; Zhuang et al., 2023; Tang et al., 2022). To ensure a more comprehensive evaluation, we also perform five-fold cross-validation on MedCL, with the results presented in Table V. Notably, our cross-validation results also significantly outperform the methods in comparison.

### .6. Scribble visualization

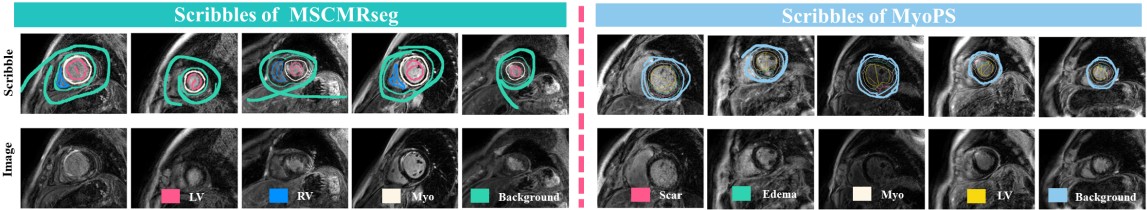

Figure I: Visualization of scribbles from MSCMRseg and MyoPS dataset.

We visualize the scribble examples from the MSCMR and MyoPS datasets. For the MSCMRseg dataset, we utilize the manual scribbles provided by (Zhang and Zhuang, 2022a), while for the MyoPS dataset, we adopt the scribbles released by (Zhang and Patel,

2024; Zhang and Zhuang, 2023). Note that for the BTCV dataset, we use full annotations instead of scribbles, aligning with previous studies (Wu et al., 2024; Chen et al., 2023; Zhou et al., 2021; Zhuang et al., 2023; Tang et al., 2022).

### .7. Pseudo code to optimize mapping $P$

We use a linear layer to implement the mapping and optimize the parameters of the linear layers to maximize the similarity between the segmentation probability map and the prototype. The optimization procedure is described in the following pseudo-code.

```
# a_t: the transpose of prototypes, d x (2m-1)
# y_hat: flatted model prediction, (2m-1) x n
# model: convnet + Mapping head
# w: the weight for regularization for smoothness

scores = torch.mm(a_t, y_hat) # prototype scores: (d x n)

with torch.no_grad():
  q = sinkhorn(scores)

p = Softmax(scores / w)
loss = - mean(q * log(p))

# Sinkhorn algorithm to compute optimal transport matrix
function sinkhorn(scores, eps=0.05, niters=3):
    P = exp(scores / eps).T              # Exponentiate and transpose the scores
    P /= sum(P)                          # Normalize P by row sum
    d, n = P.shape                       # Get the dimensions of P
    u, r, c = zeros(d), ones(d) / d, ones(n) / n  # Initialize scaling vectors

    for _ in range(niters):              # Iterate for a fixed number of iterations
        u = sum(P, dim=1)                # Update u as row sum of P
        P *= (r / u).unsqueeze(1)        # Scale P by row scaling factor
        P *= (c / sum(P, dim=0)).unsqueeze(0)  # Scale P by column scaling factor

    return (P / sum(P, dim=0, keepdim=True)).T  # Normalize and return the final result
```

For regularization, $w$ is a parameter that controls the smoothness of the mapping. We have observed that a high value of $w$, which enforces strong entropy regularization, often leads to a trivial solution where all samples collapse into a single representation and are uniformly assigned to all prototypes. Therefore, in practice, we maintain a low value for $w$. We solve the optimization using the Sinkhorn-Knopp algorithm (Cuturi, 2013). The parameter optimization during the clustering process follows the approach outlined in previous work (Caron et al., 2020).

### .8. The contribution of the unsupervised and supervised loss

We evaluate the contribution of the unsupervised and supervised loss components through two experimental setups as follows: (a) Fixed supervised loss with added unsupervised losses: As detailed in Table 4 of the manuscript, when the unsupervised losses were incorporated, while keeping the supervised loss unchanged, the average Dice score improved significantly from 0.350 to 0.833. This result demonstrates the effectiveness of the proposed unsupervised approaches. (b) Fixed unsupervised loss with varying supervised losses: As shown in Table I, the impact of the supervised loss was further examined by varying the number of scribble annotations (from 1 to 10), with and without category information. In all cases, the model incorporating unsupervised losses (denoted as MedCL-UNet) consistently outperformed the compared approaches. Meanwhile, as the number of scribble annotations increased from 1 to 10, the average Dice score on a challenging pathology segmentation task improved from 36.3% to 58.2%. Based on the results of the two experimental setups, we clarify that our proposed unsupervised losses, which constitute the primary contribution of our method, yield significant performance improvements across diverse scenarios. Furthermore, our findings indicate that when annotations are limited, the supervised losses provide essential guidance to the model in identifying the target regions of interest.

### .9. The effect of each loss terms

We provided extensive ablation studies to assess the contributions of the various loss terms in Table 4 of manuscript (Ablations of unsupervised losses), Table I of Appendix Sec 1(Ablations of supervised loss), and Table IV (detailed ablations for $L_{mix}$). We clarify the ablation details in the descriptions below: For unsupervised Loss: This component consists of $L_{mix}$, $L_{cluster}$, and $L_{ac}$. As shown in Table 4 of the manuscript. Detailed ablation results on the effects of intra-mix and inter-mix can be found in Table V. For supervised Loss: The supervised component is comprised of $L_{scribble}$ and $L_{category}$. Table I presents the baseline performance when using the combined supervised loss, while Table 1 in Appendix Section 3 provides a detailed analysis of the individual contributions of $L_{scribble}$ and $L_{category}$. Even though the contribution of $L_{category}$ is significant, the model trained without $L_{category}$ still outperforms other scribble-supervised baselines such as nnPU and nnUNet. For $L_{scribble}$, it calculate cross-entropy and Dice loss for annotated pixels as serves as the baseline. Our ablation studies highlight the crucial contribution of each loss term to the overall performance improvements.

### .10. Rationale of MedCL

We clarify that the main goal of our work is to model the anatomy distribution for better image segmentation. We use feature clustering to capture this distribution. Since medical datasets are often small and sparsely annotated, we introduce feature shuffling to generate augmented features, improving clustering quality. Regarding text prompting, this is specific to cases where we apply our method to SAM-based architectures. In such cases, feature-level shuffling involves generating diverse segmentation masks, and we use prompt sampling (via text prompts) to guide the model in producing augmented outputs that cover different classes. Thus, text prompting is used as a practical way to perform feature shuf-

fling by generating diverse segmentation masks through varied prompts, but it is not a core conceptual contribution on its own.

