# OpenReview forum: "MedCL: Learn Consistent Anatomy Distribution for Scribble-supervised Medical Image Segmentation"
_MIDL.io/2025/Conference — MIDL 2025 Poster_

### Official Review · Reviewer_6BT6 · 2025-02-21

**Confidence:** 3
**Preliminary Rating:** 2

**Summary:**

This paper presents a method to improve the training of medical image segmentation models in the context of weakly annotated datasets, for instance where only scribbles inside/outside the objects of interests are available.
The method is based on multiple components: (1) data augmentation via intra- and inter-image mixing and shuffling, (2) a text prompt encoder, (3) a prototype estimation and feature clustering.
The experiments are based on several datasets, both with regular anatomical structures and pathologies and show superior results compared to the compared baselines.

**Strengths:**

* The paper addressed an important and challenging topic, namely leveraging weak annotations.
* A number of experiments, including ablation studies, have been ran. The method seems to really outperform all other baselines on several datasets.
* The visual results seem convincing, and I particularly appreciate that the best/median/worst case have been shown.

**Weaknesses:**

* The related work and selected baselines are mostly restricted to the field of data augmentation via mixing and I am not sure I understand such a narrow focus.
  This is not at all the only way to deal with weak annotations. For instance, a number of papers in the previous MIDL conferences have introduced various approaches, for instance:
	* https://arxiv.org/abs/1805.04628
	* https://openreview.net/pdf?id=nqe6e0oJ_fL

* I am a bit confused by a number of things in the paper, especially on the feature clustering and the whole "text prompting" part, which is suddenly mentioned in Section 2.1 without any description in the Introduction.
  I could not find any example in the main paper of such prompts, so I feel like I am missing a big part of the method.
  The main paper also does not seem to include ablation studies, so it is hard to assess its importance.
  Finally, I am not sure if the baselines that the proposed method was compared to also include text prompts.

* The datasets used for the experiments are relatively small (< 50 scans)

**Detailed Comments:**

* On the feature clustering:
	* I am not sure the prototypes should be necessarily gathered in clusters. If the number of prototypes turns out to be equal to the number of clusters in the dataset structures, there would be one prototype per cluster.
	* Also, I'm not sure there should be clusters in the first place. It could just be a continuum.
	* In Equation 4, it seems the loss is minimal if the numerator is large.. so the compactness measure should actually be maximized?
	* In general, given the encoding of the features, doesn't it assume that all the structures are geometrically aligned (no extra translation/rotation)?

* Given that the main baseline is ModelMix (the paper from which most experiments are taken from and the best competing method), it would have been nice to dedicate at least a couple of sentences in the Introduction to clarify the differences with respect to this method in particular.
* The results of PuzzleMix and Co-Mixup look really bad. Can you comment on that? Are you sure that there is no problem in your implementation?
* Average and standard deviations have been reported, but it is sometimes hard to assess whether the differences are significant, for instance in Table 4. I would recommend to run some statistical tests.
* In Table 1, the left ventricle HD half of the methods are exactly the same number. Is that a typo?
* Learning only from scribbles would be indeed very nice but it seems a bit unrealistic to get such a model approved for clinical use.
  A mixed setup with both a precisely annotated subset and a subset scribble-based annotated subset seems a bit more feasible for actual use.
* A number of experiments and sometimes important pieces of information are in the supplementary material, but still referred to in the main text.
  The paper should rather be self-contained, otherwise it seems that the space limit has been circumvented.

**Justification Of The Preliminary Rating:**

While I acknowledge the effort put into the method's description, I still think some parts are a bit confusing. The paper seems to introduce three different ideas (feature shuffling, feature clustering and text prompting) but I don't really see the link between them.
I would find difficult to reproduce the method (fortunately, the authors claim they would release the code upon acceptance).

The method does seem to compare favorably to the baselines, but a more diverse sets of baselines would have been more convincing.

**Questions To Address In The Rebuttal:**

Any comment on my previous points in Weaknesses of Detailed Comments would be appreciated

---

> ### Author Response · Authors · 2025-03-06
> **Responses to Questions 11–15**
>
> **Question 11**: The results of PuzzleMix and Co-Mixup look really bad. Can you comment on that? Are you sure that there is no problem in your implementation?
>
> *Answer*: Thanks. We would like to clarify that the observed issue is due to the patch transportation strategy used by PuzzleMix and Co-Mixup, a limitation that has been noted in previous work [1]. Specifically, PuzzleMix has been shown to underperform in scribble-supervised segmentation, especially on the challenging MSCMRseg dataset, due to its patch transportation approach, which tends to distort the shape of target structures. This behavior contrasts with other mixup methods, which utilize linear interpolation or local replacement and are less likely to introduce such distortions. Co-Mixup, which also relies on a similar patch-based transportation strategy, faces the same challenges. These distortions hinder the model's ability to learn accurate shape priors, particularly when trained on small datasets, where preserving the integrity of shape information is essential for effective segmentation. We have clarified this in our revision.
>
> [1] Zhang, K., & Zhuang, X. (2022). Cyclemix: A holistic strategy for medical image segmentation from scribble supervision. In Proceedings of the IEEE/CVF Conference on Computer Vision and Pattern Recognition (pp. 11656-11665).
>
> **Question 12**: Average and standard deviations have been reported, but it is sometimes hard to assess whether the differences are significant, for instance in Table 4. I would recommend to run some statistical tests.
>
> *Answer*: Thanks. We have conducted statistical tests to evaluate the effectiveness of our proposed components and found that they have significant impact. The corresponding statistical test results have been added to Table 4 in our revised manuscript.
>
> **Question 13**: In Table 1, the left ventricle HD half of the methods are exactly the same number. Is that a typo?
>
> *Answer*: Thanks. The identical left ventricle HD values in Table 1 are not a typo. In our experiments, the left ventricle Hausdorff Distance is approximately 256 mm, which corresponds to the maximum distance given our input image dimensions (256 × 256). Since the Hausdorff Distance measures the maximum discrepancy between the predicted segmentation and the ground truth, a value of 256 mm indicates that the error reaches the boundary of the image. We followed the HD calculation procedure used in previous studies [1], and similar observations have been reported in the ModelMix manuscript (see Table 2).
>
> **Question 14**:	Learning only from scribbles would be indeed very nice but it seems a bit unrealistic to get such a model approved for clinical use. A mixed setup with both a precisely annotated subset and a subset scribble-based annotated subset seems a bit more feasible for actual use.
>
> *Answer*: Thanks. Our MedCL framework is inherently flexible and can be extended to mixed supervision settings that incorporate both precisely annotated samples and scribble-based annotations. We would like to emphasize that the core contributions of our method—specifically, the feature clustering and shuffling components—are unsupervised and independent of the annotation type.  Annotations serve as guiding cues, and we followed the experimental settings from prior scribble-supervised studies by exclusively using scribble annotations for consistency and simplicity. However, we acknowledge that incorporating a subset of precisely annotated data could further enhance performance, as demonstrated in the ablation study of CycleMix [1] (Table 4). We appreciate your suggestion and believe that a mixed supervision approach can be meaningful for clinical applications.
>
> **Question 15**: A number of experiments and sometimes important pieces of information are in the supplementary material, but still referred to in the main text. The paper should rather be self-contained, otherwise it seems that the space limit has been circumvented.
>
> *Answer*: Thanks. We would like to clarify that, due to space limitations, some experimental results and details were included in Appendix. In the revised manuscript, we have incorporated the essential information mentioned by reviewers into the main text to ensure that the paper is fully self-contained. Additional details remain available in the supplementary material for readers interested in further specifics.

---

> > ### Author Response · Authors · 2025-03-12
> >
> > We truly appreciate your time and thoughtful feedback. We have carefully addressed your concerns in our rebuttal and would be grateful if you could take a look to see if anything remains unaddressed. If any questions or uncertainties persist, we would be more than happy to provide further clarification. Thank you again for your time and consideration!

---

> ### Author Response · Authors · 2025-03-06
> **Responses to Questions 7–10**
>
> **Question 7**: Also, I'm not sure there should be clusters in the first place. It could just be a continuum.
>
> *Answer*: Thanks. Our design choice is based on the following considerations: although anatomical features can theoretically be viewed as a continuum, in practical applications, partitioning these features into discrete clusters based on classes allows us to more effectively capture anatomy distribution patterns in the data, thereby enhancing segmentation performance. Discrete clustering not only results in more compact and interpretable feature representations but also facilitates subsequent regularization and feature alignment processes. Our experimental results demonstrate that this approach offers significant advantages when working with limited and weakly annotated data. We recognize the inherent continuity of anatomical structures and will explore how to better balance continuous representations with discrete clustering in future work.
>
> **Question 8**: In Equation 4, it seems the loss is minimal if the numerator is large. so the compactness measure should actually be maximized?
>
> *Answer*: Yes, we maximize the compactness measure because it quantifies the density of the prototype distribution within clusters. By doing so, we ensure that the prototypes are tightly grouped, resulting in more compact and well-defined clusters.
>
> **Question 9**: In general, given the encoding of the features, doesn't it assume that all the structures are geometrically aligned (no extra translation/rotation)?
>
> Thanks. Our method regularizes the distribution of class-specific features so that it is not affected by translation or rotation. Specifically, we design the mapped prototypes to capture only the class distribution and coverage information by enforcing both inclusive relationships (for instance, prototypes for classes A and B can combine to form a joint prototype representing their union) and exclusive relationships (ensuring that prototypes for class A remain distinct from those for class B). These inter-class relationships reflect the underlying anatomical patterns, meaning that even if structures undergo geometric transformations, their inherent relationships remain unchanged—for example, the spatial configuration of the left ventricle, right ventricle, and myocardium forming the heart is preserved regardless of translation or rotation.
>
> **Question 10**: Given that the main baseline is ModelMix (the paper from which most experiments are taken from and the best competing method), it would have been nice to dedicate at least a couple of sentences in the Introduction to clarify the differences with respect to this method in particular.
>
> *Answer*:  Thanks. The primary difference between our method and ModelMix lies in their training paradigms. ModelMix leverages complementary tasks by training separate models on different datasets (e.g., MSCMRseg and MyoPS) and then interpolating their parameters, which allows it to incorporate knowledge from other tasks but requires extra data sources. In contrast, our approach is designed to model the anatomical distribution priors of the target segmentation task, without requiring additional data sources. We have revised the manuscript to clarify this.

---

> ### Author Response · Authors · 2025-03-06
> **Responses to Questions 3–6**
>
> **Question 3**: The main paper also does not seem to include ablation studies, so it is hard to assess its importance.
>
> *Answer*: Thanks. We provide an ablation study on the main components of our MedCL in Table 4, evaluating the effects of feature shuffling, feature clustering, and anatomy consistency regularization. The table reports results for both MedCL-SAM and MedCL-UNet, where MedCL-SAM incorporates text prompts and MedCL-UNet does not. Further ablation studies are included in the supplementary material, where we analyze the impact of text prompts (Appendix Sec. 3), batch size (Appendix Sec. 2), the effectiveness of the feature shuffling component (Appendix Sec. 4), and the role of category labels (Appendix Sec. 1).
>
> **Question 4**:  Finally, I am not sure if the baselines that the proposed method was compared to also include text prompts.
>
> *Answer*: Thanks. We would like to clarify that text prompting is specifically designed for the SAM backbone. In the ablation study presented in Table 4, we report ablation results for both MedCL-SAM and MedCL-UNet. As MedCL-SAM uses text prompts and MedCL-UNet does not, we have clarified this distinction in the revision by annotating them as MedCL-SAM (w/ text prompt) and MedCL-UNet (w/o text prompt).
>
> **Question 5**: The datasets used for the experiments are relatively small (< 50 scans)
>
> *Answer*: Thanks. We would like to clarify that our evaluations were conducted on widely recognized open datasets, including MSCMRseg, BTCV, and MyoPS. A key advantage of our approach is its ability to fine-tune large models or train lightweight convolutional models from scratch on small, weakly annotated datasets—a task that is especially challenging compared to working with large datasets. This advantage highlights the effectiveness of our proposed feature shuffling and feature clustering methods in efficiently modeling anatomical priors. Furthermore, to address concerns about dataset size, we include comprehensive 5-fold cross-validation results in Appendix Sec. 5, which demonstrate that our method significantly outperforms comparative approaches.
>
> **Question 6**: I am not sure the prototypes should be necessarily gathered in clusters. If the number of prototypes turns out to be equal to the number of clusters in the dataset structures, there would be one prototype per cluster.
>
> *Answer*: Thanks.We would like to clarify that the number of prototypes in our method is significantly greater than the number of segmentation classes. Let m denote the number of segmentation classes and n the batch size. According to our text prompt sampling procedure (described in Question 2), the total number of prototypes is given by n × (2m − 1). For example, in the MSCMRseg task with m = 3 and n = 64, we obtain 64 × (2 × 3 − 1) = 320 prototypes. These 320 prototypes are distributed as follows: 64 for each individual class (RV, MYO, LV), 64 for two-class combination, and 64 for the combination of all three classes.
>
> We first form individual clusters for each class using their corresponding 64 prototypes, enforcing similarity within each cluster to ensure compactness, while maintaining clear distinctions between clusters of different classes. Next, we regularize the anatomical prior by aligning the distribution across clusters. For instance, the 64 prototypes corresponding to the three-class combination are grouped into a large cluster that is required to be compact and closely match the aggregated representation of the individual class clusters. A similar regularization strategy is applied to the two-class combinations, where the prototypes for each pair (e.g., LV & RV, LV & MYO, MYO & RV) are clustered and regularized to form coherent groupings.

---

> ### Author Response · Authors · 2025-03-06
> **Responses to Questions 1–2**
>
> **Question 1**: The related work and selected baselines are mostly restricted to the field of data augmentation via mixing and I am not sure I understand such a narrow focus. This is not at all the only way to deal with weak annotations. For instance, a number of papers in the previous MIDL conferences have introduced various approaches, for instance:
>
> https://arxiv.org/abs/1805.04628
> https://openreview.net/pdf?id=nqe6e0oJ_fL
>
> *Answer*: Thank you for the suggestion. We have added a discussion of suggested related works in the revision.
>
> **Question 2**: I am a bit confused by a number of things in the paper, especially on the feature clustering and the whole "text prompting" part, which is suddenly mentioned in Section 2.1 without any description in the Introduction. I could not find any example in the main paper of such prompts, so I feel like I am missing a big part of the method.
>
> *Answer*: Thanks. Due to page limitations, we provide the full details of the text prompting module and its ablation studies on prompt variations in Appendix 3. In the revision, we have also added example prompts and clear references in the main text to guide readers to the appendix for more information.
>
> *Rule of text prompting*: We would like to further clarify that text prompting is a practical implementation detail specifically for applying our method to SAM-based architectures. In these cases, feature-level shuffling is achieved by generating diverse segmentation masks, and prompt sampling (through text prompts) helps ensure a variety of classes are represented in the augmented outputs. While text prompting is necessary to enable feature shuffling within prompt-driven models like SAM, it is not a central conceptual contribution of our work but rather an implementation adaptation for this specific model.
>
> *Examples of text prompts*: We provide the list of text prompts in the table below. Using the MSCMR dataset as an example, there are three classes: RV, Myo, and LV. We utilize two groups of text prompts: the noun group and the sentence description group, as shown in the table below. The ablation study comparing the performance of MedCL-SAM, SAM, and MedSAM with the two groups of prompts is presented in Table IV of the supplementary material. Our MedCL-SAM achieves promising results across various text prompts, although its performance slightly decreases when using ambiguous, longer sentence descriptions.
> | Noun              | Description                                                                                   |
> |-------------------|-----------------------------------------------------------------------------------------------|
> | RV (Right Ventricle) | Right ventricle has complex shape, triangular from the frontal aspect and crescentic from the apex. |
> | Myo (Myocardium) | Myocardium typically appears dark or black in LGE images, and has a circular shape.           |
> | LV (Left Ventricle) | Left ventricle is typically observed as a roughly elliptical or oblong structure.           |
>
> *Text prompt sampling*: We sample class combinations following the pattern below. As described in the manuscript, we first sample text prompts for each class. Taking MSCMRseg as an example, there are three foreground classes: RV, Myo, and LV. The initial sampled prompt is therefore [RV, Myo, LV]. Next, we sample combinations of these classes, with the number of classes in each combination ranging from 2 to m (where m is the total number of foreground classes). For MSCMRseg, the possible combinations are: [RV and LV, RV and Myo, Myo and LV, RV and Myo and LV]. For example, when sampling two-class combinations, we might select [RV and LV]. For three-class combinations, we get [RV and Myo and LV]. The final sampled set, in this case, could be [RV, Myo, LV, RV and LV, RV and Myo and LV], which has a dimension of 5. In general, for $m$ foreground classes, the total number of possible sampled prompts is $2m−1$ (e.g.,2×3−1=5). This progressive sampling strategy allows us to gradually expand the set of class combinations until it covers all possible subsets of the class set.
>
> *Text prompt augmentation*: Additionally, we apply data augmentation during the sampling process. For example, the conjunction "and" can be replaced with synonyms (e.g., "with", "along with"), and class names such as RV can be substituted with equivalent terms like Right Ventricle. We have revised the manuscript to provide a clearer explanation of the sampling process and included an illustrative example in Appendix Sec 3.
>
> Regarding feature clustering, we have provided detailed responses to the specific questions raised below.

---

> ### Author Response · Authors · 2025-03-06
> **Responses to Main Concerns**
>
> We sincerely thank the reviewer for acknowledging our efforts and providing valuable feedback. We have carefully addressed the concerns raised through point-by-point responses and corresponding revisions to the manuscript. We hope these changes adequately resolve the concerns. If any issues remain, please do not hesitate to let us know — we would be happy to provide further clarification as needed.
>
> **Concern 1**: The paper seems to introduce three different ideas (feature shuffling, feature clustering and text prompting) but I don't really see the link between them.
>
> Thanks. We clarify that the main goal of our work is to model the anatomy distribution for better image segmentation. We use feature clustering to capture this distribution. Since medical datasets are often small and sparsely annotated, we introduce feature shuffling to generate augmented features, improving clustering quality. Regarding text prompting, this is specific to cases where we apply our method to SAM-based architectures. In such cases, feature-level shuffling involves generating diverse segmentation masks, and we use prompt sampling (via text prompts) to guide the model in producing augmented outputs that cover different classes. Thus, text prompting is used as a practical way to perform feature shuffling by generating diverse segmentation masks through varied prompts, but it is not a core conceptual contribution on its own.
>
> In short, feature clustering is the core, feature shuffling supports it through augmentation, and text prompting is an implementation detail of feature shuffling for SAM-based setups. We have clarified this connection in the revised paper and elaborate on it in the Appendix Sec 9.
>
> **Concern 2**: I would find difficult to reproduce the method (fortunately, the authors claim they would release the code upon acceptance).
>
> Thanks. We would like to clarify that our method is both reproducible and generalizable to various tasks. It can also be easily integrated as a modular component for different backbones, such as UNet, nnUNet, and SAM. To ensure the reproducibility of our work, we will  provide the code for our implementations across these backbones upon acceptance. Regarding the text prompt sampling and feature clustering components, we have included detailed explanations in our responses to the specific questions below.

---

### Official Review · Reviewer_oAKd · 2025-02-22

**Confidence:** 4
**Preliminary Rating:** 5
**Final Rating:** 5

**Summary:**

This paper presents MedCL, a new framework for medical image segmentation that uses weak scribble annotations instead of full labels, reducing the time and effort required for large-scale manual labeling. The key ideas proposed include the use of mix-up techniques to learn compact, distinctive anatomical patterns. By combining this unsupervised anatomy learning with a small amount of scribble supervision, the framework can accurately segment both regular organs and irregular pathologies, outperforming many existing methods. SAM and U-Net were also implemented using this framework.

**Strengths:**

The strengths of this paper can be summarized as follows:
1. The overall writing quality, tables, and figure visualizations are good.
2. The complexity of the model designs and the inclusion of foundation models like SAM is innovative for the medical imaging domain.
3. The presented experiments in the results section of this paper are thorough.

**Weaknesses:**

One thing I was considering is the contribution between loss of unsupervised part and supervised part. Also, are there more ablation studies on the contribution of different loss terms on the results (beside Lac and Lcluster)?

**Detailed Comments:**

One thing I was considering is the contribution between loss of unsupervised part and supervised part. Also, are there more ablation studies on the contribution of different loss terms on the results (beside Lac and Lcluster)?

**Justification Of The Final Rating:**

Based on the author's response, myy questions are addressed. The problem of using scribbles for annotation can make the annotation process easier in the real-life context of pathologists. The ablation studies are thorough as well.

**Justification Of The Preliminary Rating:**

My rating is based on the overall structure, ideas, and visualization of this paper, all of which are good. The problem of using scribbles for annotation can make the annotation process easier in the real-life context of pathologists. The ablation studies are thorough as well.

**Questions To Address In The Rebuttal:**

1. One thing I was considering is the contribution between loss of unsupervised part and supervised part.
2. Also, are there more ablation studies on the contribution of different loss terms on the results (beside Lac and Lcluster)?

---

> ### Author Response · Authors · 2025-03-07
>
> We appreciate the reviewer’s recognition of the quality of our writing, the innovation of our approach, and the thoroughness of our evaluation. The recommendation is greatly appreciated, and detailed responses to the specific comments are provided below.
>
> **Question 1**: One thing I was considering is the contribution between loss of unsupervised part and supervised part.
>
> *Answer*: Thanks. We evaluate the contribution of the unsupervised and supervised loss components through two experimental setups as follows:
>
> (a) Fixed supervised loss with added unsupervised losses:
> As detailed in Table 4 of the manuscript, when the unsupervised losses were incorporated,while keeping the supervised loss unchanged, the average Dice score improved significantly from 0.350 to 0.833. This result demonstrates the effectiveness of the proposed unsupervised approaches.
>
> (b) Fixed unsupervised loss with varying supervised losses:
> As shown in Table I of Appendix Section 1, the impact of the supervised loss was further examined by varying the number of scribble annotations (from 1 to 10), with and without category information. In all cases, the model incorporating unsupervised losses (denoted as MedCL-UNet) consistently outperformed the compared approaches. Meanwhile, as the number of scribble annotations increased from 1 to 10, the average Dice score on a challenging pathology segmentation task improved from 36.3% to 58.2%.
>
> Based on the results of the two experimental setups, we would like to clarify that our proposed unsupervised losses, which constitute the primary contribution of our method, yield significant performance improvements across diverse scenarios. Furthermore, our findings indicate that when annotations are limited, the supervised losses provide essential guidance to the model in identifying the target regions of interest. We have added the clarification about the contribution between loss of unsupervised part and supervised part in Appendix Sec 8 of revision.
>
> **Question 2**: Also, are there more ablation studies on the contribution of different loss terms on the results (beside Lac and Lcluster)?
>
> *Answer*: Thanks. We provided extensive ablation studies to assess the contributions of the various loss terms beyond $L_{ac}$ and $L_{cluster}$ in Table 4 of manuscript (Ablations of unsupervised losses), Table I of Appendix Sec 1(Ablations of supervised loss), and Table IV (detailed ablations for $L_{mix}$). We clarify the ablation details in the descriptions below:
>
> For unsupervised Loss:  This component consists of $L_{mix}$, $L_{cluster}$, and $L_{ac}$. As shown in Table 4 of the manuscript, incorporating $L_{mix}$ (implemented via feature shuffling) significantly improves performance compared to using only the supervised loss. Specifically, the average Dice score increases from 35.0% to 52.1% on MedCL-SAM (with text prompt) and from 22.2\% to 56.3\% on MedCL-UNet (without text prompt). Detailed ablation results on the effects of intra-mix and inter-mix can be found in Table IV of Appendix Section 4.  We have further clarified this in our revision.
>
> For supervised Loss:  The supervised component is comprised of $L_{scribble}$ and $L_{category}$. Table 4 presents the baseline performance when using the combined supervised loss, while Table 1 in Appendix Section 3 provides a detailed analysis of the individual contributions of $L_{scribble}$ and $L_{category}$. Even though the contribution of $L_{category}$ is significant, the model trained without $L_{category}$ still outperforms other scribble-supervised baselines such as nnPU and nnUNet.
> For $L_{scribble}$, it calculate cross-entropy and Dice loss for annotated pixels as serves as the baseline.
>
> We have added a discussion to summarize the effect of each loss term in revised Appendix Sec 9. Our ablation studies highlight the crucial contribution of each loss term to the overall performance improvements.

---

> > ### Comment · Reviewer_oAKd · 2025-03-14
> >
> > Thanks for the detailed response from the authors. I have no concern based on this response.

---

### Official Review · Reviewer_3iym · 2025-02-28

**Confidence:** 4
**Preliminary Rating:** 4
**Recommendation:** Oral

**Summary:**

The paper presents a novel MedCL framework to learn the anatomy distribution for medical image segmentation. The implementation is based on SAM and UNet architectures.

**Strengths:**

The paper explores feature shuffling and inter- and intra-image operations to learn anatomy priors and also introduces a feature clustering approach to understanding the inherent anatomy relationships. Evaluated multiple datasets to test the framework robustly and improved performance using the scribble-supervised clustering-based framework (MedCL) method. Also, the technique combined with weak supervision helps segment irregular pathologies, which is challenging.

**Weaknesses:**

It would have been a good idea to train the models on a color dataset rather than only on low-contrast images. Incorporating color information could provide richer features, potentially enhancing the model's ability to distinguish structures and improving overall performance. The proposed algorithm might have benefited from this additional information, offering a more comprehensive evaluation.

**Detailed Comments:**

The paper explores feature shuffling and inter- and intra-image operations to learn anatomy priors and also introduces a feature clustering approach to understanding the inherent anatomy relationships. Evaluated multiple datasets to test the framework robustly and improved performance using the scribble-supervised clustering-based framework (MedCL) method. Also, the technique combined with weak supervision helps segment irregular pathologies, which is challenging.

**Justification Of The Preliminary Rating:**

The paper explores feature shuffling and inter- and intra-image operations to learn anatomy priors and also introduces a feature clustering approach to understanding the inherent anatomy relationships. Evaluated multiple datasets to test the framework robustly and improved performance using the scribble-supervised clustering-based framework (MedCL) method. Also, the technique combined with weak supervision helps segment irregular pathologies, which is challenging. Considering all these strengths I would go with weak accept.

**Questions To Address In The Rebuttal:**

NA

**Special Issue:**

Yes

---

> ### Author Response · Authors · 2025-03-07
>
> We sincerely thank the reviewer for acknowledging the novelty of our method. We truly appreciate the recommendation and provide detailed responses to the specific comments below.
>
> **Question 1**: It would have been a good idea to train the models on a color dataset rather than only on low-contrast images. Incorporating color information could provide richer features, potentially enhancing the model's ability to distinguish structures and improving overall performance. The proposed algorithm might have benefited from this additional information, offering a more comprehensive evaluation.
>
> *Answer*: Thanks for your comment! We agree that incorporating color information could provide richer features and potentially enhance the model’s ability to distinguish structures. While our current work focuses on low-contrast images, our algorithm can be directly extended to color datasets, and we plan to explore this direction in our future work.

---

### Official Review · Reviewer_2j9h · 2025-03-01

**Confidence:** 5
**Preliminary Rating:** 2
**Recommendation:** Poster
**Final Rating:** 4

**Summary:**

Authors present a method for weakly supervised image segmentation
using scribble inputs, and text and bounding-box inputs in one
instance. The main contribution is the training strategy. Authors
include an elaborate random shuffling strategy for augmentation, which
is based on mix-up, and a regularization strategy that combines multiple
segmentations as one and enforces consistency across individuals and
combinations. Further, segmentations are converted into prototypes and
a similar consistency is enforced in the prototype space. Lastly, a
clustering is enforced to construct the prototypes and also clustering
of the segmentations. Experiments with three different datasets are
presented. Comparisons with multiple methods are made.

**Strengths:**

+ There are at least two technical contributions, which are novel to
  the best of my knowledge and interesting in my opinion. I find the
  consistency between individual segmentations / prototypes and their
  combinations especially interesting. Use of mix-up with text-prompts
  is also interesting.
+ Authors spent a great amount of effort for evaluations with three
  different datasets and an ablation study.
+ Results are quite interesting. It is difficult to outperform a well
  designed UNet, which the nnUnet heuristic provides. It seems here
  authors is able to outperform this baseline on different datasets.

**Weaknesses:**

+ The explanation of the methods requires substantial amount of
  improvement, it is really not what one expects from a top-level
  article. It is also unfortunate because I think the paper is nice
  but the lower quality of the writing shadows the higher quality of
  the content (as far as I can infer from the writing). Specifically,
  - The word 'shuffle' is not the best one to describe the method. I
    recommend simply using mix-up or mixing.
  - What are the different text prompts used? how do the authors
    sample them?
  - How is the union operation defined over the segmentation labels?
    Are these labels binary maps, probabilities, categorical labels?
  - How is the addition defined for segmentations? This appears before
    Equation 1 and then in Equation 5.
  - What is $\textbf{y}'$? This has not been defined.
  - In the online mapping part, why is the prediction of size $m\times
    h\times w$? It was of size $(2m-1) \times h \times w$ in the
    previous section.
  - The mapping $P$ needs to be better explained, especially
    justifying the regularization type used.
  - Is the mapping P working on binary maps, one-hot encodings,
    probabilities or multi-label maps?
  - Is Equation 2 optimized over a dataset or over a sample. If a
    dataset, is Equation 3 still the solution?
  - The section on Weak Supervision starts by saying MedCL is
    conducted in a unsupervised setting, yet
    $\mathcal{L}_{\textrm{mix}}$ seems to use ground truth
    segmentations. Is there an inconsistency here?
+ The explanation of the experiments is slightly better than the
  methods but still require much work. Specifically,
  - What do the scribbles look like? The paper should be
    self-contained.
  - Do authors use scribbles for BTCV?
  - Do authors use SAM or UNet backbone for Table 2 and Table 3?
  - The design of the supervision sensitivity experiment is not
    clear. Why does the accuracy of a fully supervised model decrease?

**Detailed Comments:**

Please see above.

**Justification Of The Final Rating:**

This is a borderline case in my opinion. The idea is nice and authors have spent quite a bit of time in the rebuttal, and provided good answers.
The paper, however, had to change quite a bit. If the amount of changes is an important point, then AC should take that into account while deciding.

**Justification Of The Preliminary Rating:**

The approach has interesting contributions and results are favoring
the model. However, the explanations in the methods section and the
experiments section are not good for a high-quality article. I cannot
propose acceptance of the paper in this state even though I believe it
is a nice idea. Depending on how much modification is allowed, perhaps
authors can update the paper - with substantial modifications in the
explanation - and then submit in the rebuttal. However, the ACs and
PCs should decide how much change is allowed. I think the writing
requires substantial changes.

**Questions To Address In The Rebuttal:**

Please see above.

**Special Issue:**

No

---

> ### Author Response · Authors · 2025-03-05
> **Responses to Questions 11–16**
>
> **Question 11**: The section on weak supervision starts by saying MedCL is conducted in a unsupervised setting, yet $L_{mix}$ seems to use ground truth segmentations. Is there an inconsistency here?
>
> *Answer*: Thanks. We would like to highlight that L_mix uses the model's predictions for regularization and does not involve any ground truth segmentation. The Equation 1 is formulated as follows:
>
> $L_{mix} = sim(\hat{y}_{12}, M(\hat{y}'_1, \hat{y}'_2)),$
>
> where the $\hat{y}$ notation indicates that the segmentation used is the model's
> prediction, not the ground truth.  We have revised the manuscript to further clarify this.
>
> **Question 12**: The explanation of the experiments is slightly better than the methods but still require much work.
>
> *Answer*: Thanks. We have provided point-by-point responses and revised the manuscript to address your concerns.
>
> **Question 13**: What do the scribbles look like? The paper should be self-contained.
>
> *Answer*: Thanks. In the original manuscript, we included the scribble examples in Appendix 6 due to page limitations. In the revised manuscript, we have incorporated the scribble illustrations within the main text to make the paper more self-contained, and provided additional examples in Appendix 6.
>
> **Question 14**: Do authors use scribbles for BTCV?
>
> *Answer*: Thanks. We clarify that we do not use scribbles in order for consistency with previous works like [4]. We follow the same data split as previous methods, which use full annotations instead of scribbles. This has been clarified in the revised manuscript.
>
> [4]  Linshan Wu, Jiaxin Zhuang, and Hao Chen. Voco: A simple-yet-effective volume contrastive
> learning framework for 3d medical image analysis. In Proceedings of the IEEE/CVF
> Conference on Computer Vision and Pattern Recognition, pages 22873–22882, 2024.
>
> **Question 15**: Do authors use SAM or UNet backbone for Table 2 and Table 3?
>
> *Answer*: Thanks. For Table 3, we evaluate both the SAM and UNet backbones separately, with their results highlighted in pink for clarity.  Table 2 (BTCV), we use nnUNet backbone,  as described on page 5 in the preprocessing section. We have revised the manuscript to include this description in the table as well.
>
> **Question 16**: The design of the supervision sensitivity experiment is not clear. Why does the accuracy of a fully supervised model decrease?
>
> *Answer*: Thanks. We would like to clarify that, in the supervision sensitivity ablation study, the performance of all compared methods increases and gradually converges as the number of scribble annotations increases. We would like to point out that the first two subfigures of Figure 4 illustrate the impact of supervision amount; as the annotations increase, the model's performance improves as well. The last two subfigures show *the impact of bounding box shift*; as the bounding box shift increases, the performance of SAM and MedSAM decreases significantly, while our model maintains robust performance.

---

> > ### Author Response · Authors · 2025-03-05
> >
> > We have carefully addressed your concerns in our rebuttal and would greatly appreciate it if you could take a look to see if any of your concerns remain. If there are any remaining issues or aspects that need further clarification, we would be happy to provide detailed responses. We truly appreciate your time and consideration. Please let us know if there is anything we can further clarify.

---

> ### Author Response · Authors · 2025-03-05
> **Responses to Questions 6–10**
>
> **Question 6:** What is y′? This has not been defined.
>
> *Answer*: Thanks. The symbol y' defines the mixed segmentation. In Section 3.1 (Mixing Features), we introduce the superscript ' as the notation for mixing, where x' represents the mixed image and y' denotes the mixed segmentation. This has been clarified in the revised manuscript.
>
> **Question 7**: In the online mapping part, why is the prediction of size m×h×w? It was of size (2m−1)×h×w in the previous section.
>
> *Answer*: We apologize for the typo, the prediction size is (2m−1)×h×w. We have revised it in the manuscript.
>
> **Question 8**: The mapping P needs to be better explained, especially justifying the regularization type used.
>
> *Answer*: Thanks. We use an linear layer to implement the mapping $P$ and optimize the parameters of the linear layers to maximize the similarity between the segmentation probability map and the prototype. The optimization procedure is described in the following pseudo-code.
>
> ```pseudo
> # a_t: the transpose of prototypes (d x m)
> # y_hat: flatted model prediction: (m x n)
> # model: convnet + Mapping head
> # w: the weight for regularization for smoothness
>
> scores = torch.mm(a_t, y_hat) # prototype scores: (d x n)
>
> with torch.no_grad():
>   q = sinkhorn(scores)
>
> p = Softmax(scores / w)
> loss = - mean(q * log(p))
>
> # Sinkhorn algorithm to compute optimal transport matrix
> function sinkhorn(scores, eps=0.05, niters=3):
>     P = exp(scores / eps).T            # Exponentiate and transpose the scores
>     P /= sum(P)                        # Normalize P by row sum
>     d, n = P.shape                     # Get the dimensions of P
>     u, r, c = zeros(d), ones(d) / d, ones(n) / n  # Initialize scaling vectors
>
>     for _ in range(niters):            # Iterate for a fixed number of iterations
>         u = sum(P, dim=1)              # Update u as row sum of P
>         P *= (r / u).unsqueeze(1)      # Scale P by row scaling factor
>         P *= (c / sum(P, dim=0)).unsqueeze(0)  # Scale P by column scaling factor
>
>     return (P / sum(P, dim=0, keepdim=True)).T  # Normalize and return the final result
> ```
> For regularization, $w$ is a parameter that controls the smoothness of the mapping. We have observed that a high value of $w$, which enforces strong entropy regularization, often leads to a trivial solution where all samples collapse into a single representation and are uniformly assigned to all prototypes. Therefore, in practice, we maintain a low value for $w$. We solve the optimization using the Sinkhorn-Knopp algorithm [3]. The parameter optimization during the clustering process follows the approach outlined in previous work [1, 2]. We further clarified this in the revised manuscript and provide the pseudo code of optimization implemention in the Appendix.
>
> [1] Asano, Y.M., Rupprecht, C., Vedaldi, A.: Self-labelling via simultaneous clustering and representation learning. International Conference on Learning Representations (ICLR) (2020)
>
> [2] Caron, Mathilde, et al. "Unsupervised learning of visual features by contrasting cluster assignments." Advances in neural information processing systems (2020).
>
> [3] Marco Cuturi. Sinkhorn distances: Lightspeed computation of optimal transport. Advances in neural information processing systems (NIPS), 2013.
>
> **Question 9**: Is the mapping P working on binary maps, one-hot encodings, probabilities or multi-label maps?
>
> *Answer*: Thanks. The mapping P working on softmax-generated multi-label probability maps of predicted segmentation. We have clarified this in the manuscript.
>
> **Question 10**: Is Equation 2 optimized over a dataset or over a sample. If a dataset, is Equation 3 still the solution?
>
> *Answer*: Thanks. We optimize Equation 2 over a batch and perform an ablation study in Appendix 2. Our results show a significant improvement as the batch size increases from 16 to 64. While the ideal scenario would involve using a larger batch size, we are constrained by available computational resources, so we use a batch size of 64. If optimization were to be performed over the entire dataset, we would need to save the embeddings, update and aggregate them. This can be achieved with an adaptation, and we would like to leave it as a future work.

---

> ### Author Response · Authors · 2025-03-05
> **Responses to Questions 1–5**
>
> **Question 1**: The explanation of the methods requires substantial amount of improvement, it is really not what one expects from a top-level article. It is also unfortunate because I think the paper is nice but the lower quality of the writing shadows the higher quality of the content (as far as I can infer from the writing).
>
> *Answer*: We thank the reviewer for recognizing that our paper is nice and for the valuable suggestions. We have carefully revised the manuscript to improve the clarity and completeness of the method descriptions, addressing the writing issues raised. We hope these improvements better reflect the quality of our work.
>
>
> **Question 2**: The word 'shuffle' is not the best one to describe the method. I recommend simply using mix-up or mixing.
>
> *Answer*: Thanks. We have replaced the description of ‘shuffle’ to “mixing” in the revision of manuscript, which has been attached to our rebuttal.
>
> **Question 3**: What are the different text prompts used? how do the authors sample them?
>
> *Answer*:  Thanks. We provide the list of text prompts in the table below. Using the MSCMR dataset as an example, there are three classes: RV, Myo, and LV. We utilize two groups of text prompts: the noun group and the sentence description group, as shown in the table below. The ablation study comparing the performance of MedCL-SAM, SAM, and MedSAM with the two groups of prompts is presented in Table IV of the supplementary material. Our MedCL-SAM achieves promising results across various text prompts, although its performance slightly decreases when using ambiguous, longer sentence descriptions.
> | Noun              | Description                                                                                   |
> |-------------------|-----------------------------------------------------------------------------------------------|
> | RV (Right Ventricle) | Right ventricle has complex shape, triangular from the frontal aspect and crescentic from the apex. |
> | Myo (Myocardium) | Myocardium typically appears dark or black in LGE images, and has a circular shape.           |
> | LV (Left Ventricle) | Left ventricle is typically observed as a roughly elliptical or oblong structure.           |
>
> We sample class combinations following the pattern below. As described in the manuscript, we first sample text prompts for each class. Taking MSCMRseg as an example, there are three foreground classes: RV, Myo, and LV. The initial sampled prompt is therefore [RV, Myo, LV]. Next, we sample combinations of these classes, with the number of classes in each combination ranging from 2 to m (where m is the total number of foreground classes). For MSCMRseg, the possible combinations are: [RV and LV, RV and Myo, Myo and LV, RV and Myo and LV]. For example, when sampling two-class combinations, we might select [RV and LV]. For three-class combinations, we get [RV and Myo and LV]. The final sampled set, in this case, could be [RV, Myo, LV, RV and LV, RV and Myo and LV], which has a dimension of 5. In general, for $m$ foreground classes, the total number of possible sampled prompts is $2m−1$ (e.g.,2×3−1=5). This progressive sampling strategy allows us to gradually expand the set of class combinations until it covers all possible subsets of the class set.
>
> Additionally, we apply data augmentation during the sampling process. For example, the conjunction "and" can be replaced with synonyms (e.g., "with", "along with"), and class names such as RV can be substituted with equivalent terms like Right Ventricle. We have revised the manuscript to provide a clearer explanation of the sampling process and included an illustrative example in the supplementary material.
>
> **Question 4**: How is the union operation defined over the segmentation labels? Are these labels binary maps, probabilities, categorical labels?
>
> *Answer*: Thanks, the union operation defines the total area covered by the segmentation labels. We perform pixel-wise addition of the softmax-generated probability maps corresponding to the segmentation labels. These labels are represented as probabilities, and the resulting output is a combined probability map over the segmentation labels. We have clarified this in the revision.
>
> **Question 5**: How is the addition defined for segmentations? This appears before Equation 1 and then in Equation 5.
>
> *Answer*: Thanks. The addition refers to pixel-wise addition of the probability maps. In Equation 1, we apply this operation to combine the weighted segmentation probability maps, implementing the mixup technique. In Equation 5, the addition represents the union operation, which calculates the total area covered by the segmentation labels, as explained in Answer 4. We have clarified this process in the revised manuscript.

---

> ### Author Response · Authors · 2025-03-05
> **Responses to Main Concerns**
>
> We sincerely thank the reviewer for recognizing the quality of our work and for the detailed comments about paper writing. We have provided point-by-point responses and revised the manuscript to address the writing issues raised. We hope that the revisions and our responses effectively address the concerns raised. However, if any concerns remain, please feel free to point them out, and we will promptly provide a detailed response to resolve them.
>
> **Clarification of Modifications**: We would like to clarify that we have made every effort to keep the modifications minimal while addressing all the writing issues raised by the reviewer. These changes are solely aimed at improving clarity and do not alter the core ideas or contributions of our work. The revised manuscript serves as a reference for these updates.

---

### Author Rebuttal · Authors · 2025-03-08

**Rebuttal:**

We sincerely thank the reviewers for their constructive feedback. We have provided our point-by-point responses and an updated manuscript with revisions clearly marked in colored fonts. We appreciate the positive remarks on our novel idea (R1, R2, R3), thorough evaluations (R1, R2, R3, R4), and writing quality (R3). We have addressed the writing issues raised by R1 and clarified the confusing parts noted by R3. Our key revisions include:

1)Feature Clustering: Added pseudocode to clarify the optimization and regularization process of prototype mapping (Appendix Sec. 7).

2)Text Prompting: Included more text prompt examples (Appendix Sec. 3), an illustration of text prompt sampling (Appendix Sec. 3), and a discussion about the connection between text prompting and feature shuffling (Appendix Sec. 10).

3)Scribble Examples: Added scribble visualizations in the manuscript and provided additional examples in Appendix Sec. 6.

If there are any further concerns, please feel free to let us know, and we will gladly provide further clarification.

**Supporting Material:**

/attachment/c77bf69d16be3c44889baf736847580d6a3f8935.pdf

---

### Comment · Area_Chair_QdAC · 2025-03-09
**Discussion Period**

Dear Reviewers,​

Thanks for your time and effort in reviewing this paper. This is the right time to discuss this paper with each other.​

The authors have provided a rebuttal to your comments and uploaded a revision. Please review their responses and the revised manuscript. For the preliminary recommendation, we have clear diverse opinions, that is, two weak rejects, one weak accept, and one strong accept.​ Considering the authors' responses and the discussion, please update your rating and assessments for the paper.

Any discussion is welcome, and you may consider reading each other's reviews, posting questions for clarification, and reaching a consensus.​

Best,
Your AC

---

### Comment · Area_Chair_QdAC · 2025-03-14
**Urgent discussion due in about one day**

Dear all the Reviewers,

The discussion period is nearing its conclusion. Please update your final rating with justification. We still have diverse ratings: two weak rejects, one weak accept, and one strong accept. Any discussion is welcome!


Best, \\\
Your AC

---

### Meta-Review · Area_Chair_QdAC · 2025-03-22

**Recommendation:** Accept (Poster)
**Confidence:** 4

**Metareview:**

Three of the four reviewers support accepting this work and believe it will generate meaningful discussions. I believe the rebuttal has addressed most of the important comments. The authors are requested to revise the camera-ready version in accordance with the feedback provided in the rebuttal.